# Ideas and perspectives: Tracing terrestrial ecosystem water fluxes using hydrogen and oxygen stable isotopes – challenges and opportunities from an interdisciplinary perspective

Daniele Penna[1], Luisa Hopp[2], Francesca Scandellari[3], Scott T. Allen[4], Paolo Benettin[5], Matthias Beyer[6], Josie Geris[7], Julian Klaus[8], John D. Marshall[9], Luitgard Schwendenmann[10], Till H.M. Volkmann[11], Jana von Freyberg[4,12], Anam Amin[13], Natalie Ceperley[14], Michael Engel[3], Jay Frentress[3], Yamuna Giambastiani[1], Jeff J. McDonnell[15], Giulia Zuecco[12], Pilar Llorens[16], Rolf T.W. Siegwolf[17], Todd E. Dawson[18], James W. Kirchner[4,12]

[1]Department of Agricultural, Food and Forestry Systems, University of Florence, Florence-Firenze, Italy
[2]Department of Hydrology, University of Bayreuth, Bayreuth, Germany
[3]Faculty of Science and Technology, Free University of Bozen-Bolzano, Bozen-Bolzano, Italy
[4]Department of Environmental Systems Science, ETH Zürich, Zurich, Switzerland
[5]Laboratory of Ecohydrology ENAC/IIE/ECHO, EPFL, Lausanne, Switzerland
[6]Federal Institute for Geosciences and Natural Resources (BGR), Hannover, Germany
[7]School of Geosciences, University of Aberdeen, Aberdeen, United Kingdom
[8]Catchment and Eco-Hydrology research group, Luxembourg Institute of Science and Technology (LIST), Esch-sur-Alzette, Luxembourg
[9]Department of Forest Ecology and Management, Swedish University of Agricultural Sciences, Umeå, Sweden
[10]School of Environment, The University of Auckland, Auckland, New Zealand
[11]Biosphere 2 Earth Science, University of Arizona, USA
[12]Swiss Federal Research Institute WSL, Birmensdorf, Switzerland
[13]Department of Land, Environment, agriculture and Forestry, University of Padua, Padua-Padova, Italy
[14]Institute of Earth Surface Dynamics, Faculty of Geosciences and Environment, University of Lausanne, Lausanne, Switzerland
[15]Global Institute for Water Security and School of Environment and Sustainability, University of Saskatchewan, Saskatoon, Canada
[16]Institute of Environmental Assessment and Water Research (IDAEA-CSIC), Barcelona, Spain
[17]Lab of Atmospheric Chemistry, Paul Scherrer Institute (PSI), Villigen, Switzerland
[18]Center for Stable Isotope Biogeochemistry, University of California – Berkeley, Berkeley, USA

*Correspondence to*: Francesca Scandellari, Free University of Bozen-Bolzano, Faculty of Science and Technology, piazza Università 5, 39100 Bozen-Bolzano, Italy (francesca.scandellari@unibz.it)

**Abstract.** In this commentary, we summarize and build upon discussions that emerged during the workshop "Isotope-based
studies of water partitioning and plant-soil interactions in forested and agricultural environments" held in San Casciano Val di Pesa, Italy, in September 2017. Quantifying and understanding how water cycles through the Earth's critical zone is important to provide society and policy makers with the scientific background to manage water resources sustainably,

especially considering the ever-increasing worldwide concern about water scarcity. Stable isotopes of hydrogen and oxygen in water have proven to be a powerful tool to track water fluxes in the critical zone. However, both mechanistic complexities
(e.g., mixing and fractionation processes, heterogeneity of natural systems) and methodological issues (e.g., lack of standard protocols to sample specific compartments, such as soil water and xylem water) limit the application of stable water isotopes in critical zone science. In this commentary, we examine some of the opportunities and critical challenges of isotope-based ecohydrological applications, and outline new perspectives focused on interdisciplinary research opportunities for this important tool in water and environmental science.

## 1 Understanding water availability in the environment

Understanding water fluxes in the critical zone, the thin dynamic skin of the Earth that extends from the top of the vegetation canopy, through the soil, down to groundwater (Brooks et al., 2015), is becoming increasingly important as the climate changes, as human population grows, and as water supplies become increasingly constrained (OECD, 2012; WWAP, 2015). Although human water use often relies on rivers or aquifers, these resources are maintained by critical zone processes that
determine the movement of water downward to groundwater, lakes and streams ("blue water"), or upward to the atmosphere via evapotranspiration ("green water"). A better understanding of the factors that control the availability and the fate of water in the critical zone is vital to maintaining ecosystem services in a changing world (Grant and Dietrich, 2017). A more detailed mechanistic understanding of water fluxes in the critical zone would serve at least two important purposes: first, it would enable hydrological and climate models to better predict changes in green and blue water fluxes; second, it would
support management and conservation strategies that promote long-term sustainability of water resources and related ecosystem functions.

Given the variety of intertwined processes at work in the critical zone, understanding water movement through terrestrial ecosystems is inherently interdisciplinary. Critical zone processes have often been examined separately within different
disciplines, such as hydrology, soil physics, forest and landscape ecology, agroecology, biogeochemistry, and plant physiology. Stable isotopes of hydrogen ($^2$H) and oxygen ($^{18}$O) are effective tools for tracing water movement through soils, aquifers, streams, plants, and the atmosphere, and therefore can connect the disciplines mentioned above. These stable isotopes have been used as hydrological and ecophysiological tracers for more than five decades (Kendall and McDonnell, 1998; Vitvar et al, 2005; Dawson et al., 2002; Werner et al., 2012). Advancements in isotope-based tools and methods (e.g.,
Volkmann et al., 2014, 2016a; von Freyberg et al., 2017; see also the review by Sprenger et al., 2015 for pore water analysis) have recently contributed to interdisciplinary research on critical-zone water movement. A search in any literature database will reveal that there has been a sharp increase in the number of papers published on these topics in the last 10-15 years, corresponding to the introduction of commercially available and affordable laser spectroscopy systems. Laser spectrometers allow for simultaneous analysis of hydrogen and oxygen isotopes, and are cheaper and easier to operate than isotope-ratio

mass spectrometers. This has made it easier than before to analyse isotopic data in the dual-isotope space ($\delta^{18}O$ vs. $\delta^2H$). Consequently, recent studies have revealed problems in the simplifying assumptions that underlie past investigations, especially those related to steady-state and well-mixed conditions, i.e. the assumption that water in the subsurface mixes instantaneously and completely in one common reservoir, so that no differences in isotopic composition would be observed in the subsurface.


These topics were intensively discussed at the recent workshop on "Isotope-based studies of water partitioning and plant-soil interactions in forested and agricultural environments" held at Villa Montepaldi, San Casciano in Val di Pesa, Florence, Italy on 27-29 September 2017. The workshop brought together scientists from 12 countries who use stable isotopes of hydrogen and oxygen to study water movement across the critical zone. The objective was to share perspectives on major obstacles

(and their potential solutions) in applying isotope analyses in critical-zone studies. This paper reports the main outcomes of the workshop, summarizing perspectives on several urgent challenges and future research opportunities.

**2 Stable isotopes of hydrogen and oxygen: versatile and interdisciplinary tools**

Several stable isotopes of hydrogen and oxygen are naturally present in the water molecule, allowing for effective tracing of water as it moves through the critical zone. In general, we have good theoretical knowledge about individual chemical,

physical and biological processes that control the isotopic composition of water (Gat and Gonfiantini, 1981; Kendall and McDonnell, 1998). Specifically, the interactions between the vapour, liquid, and solid phases of water explain most of their isotope variability. Applying this theoretical knowledge to real-world conditions, models have been developed to explain the isotopic composition of (liquid) precipitation condensing from cloud vapour (Dansgaard, 1964; Gat, 1980; Gat, 1996; Clark and Fritz, 1997). Although these models were shown to yield reliable predictions at annual time scales, predicting the

isotopic composition of water in the atmosphere on shorter time scales remains difficult due to its short residence time (around 9 days; van der Ent and Tuinenburg, 2017) and non-uniform atmospheric mixing. Other models are available to explain the isotopic fractionation that occurs during evaporation from water bodies (Craig and Gordon, 1965), from the upper part of the soil profile (e.g., Barnes and Allison, 1988), from plant canopies (Cernusak et al., 2016; Allen et al., 2017) and even within the soil (Lin and Horita 2016). Predicting fractionation from water bodies is relatively easy because they are

reasonably well mixed near the surface, while predicting the isotopic composition of water in soils and canopies remains difficult (Sprenger et al., 2016).

It is usually assumed that root water uptake does not alter the isotopic composition of the water in the roots or the stem, and therefore xylem water samples from plant tissue should reflect the isotopic composition of the source water (Dawson et al.,

2002, and references cited therein). However, a few studies have suggested that plants, particularly halophytes and xerophytes, may fractionate the water they are using (Ellsworth and Williams, 2007), which results in an enrichment of

heavy isotopes in the surrounding soil water under certain conditions (Vargas et al., 2017). At the leaf level, evaporation during transpiration can cause strong isotopic enrichment of the heavier isotopes $^2$H and $^{18}$O in the leaf water that remains (Dawson and Ehleringer, 1998).

Recent technological advancements can help gather isotope measurements at higher temporal or spatial resolution. The oldest, but still most common device for analysing stable isotopes of light elements is the isotope-ratio mass spectrometer (IRMS) coupled with different peripherals allowing different sample media to be processed. In the past 10-15 years, new types of isotope analysers have become widely available, based on the use of tuneable diode lasers that can scan across a range of frequencies (off-axis integrated cavity output spectroscopy, OA-ICOS, and cavity ring-down spectroscopy, CRDS). The repeatability and reproducibility of these instruments are comparable to IRMS (Penna et al., 2010; 2012), but they are substantially cheaper and can be installed directly in the field for continuous, automatic measurements of liquid or water vapour samples (e.g., Berman et al., 2009; Pangle et al., 2013; Oerter et al., 2017; von Freyberg et al., 2017). However, laser instruments are sensitive to interference by organic substances that are often present in plant and soil water samples, and also to background gas compositional changes (in the case of cavity ring-down spectroscopy, see Gralher et al., 2018). Recent efforts have therefore been directed towards identifying the interfering molecules and providing sample preparation protocols and software tools to avoid or correct for such interferences (e.g., West et al., 2011; Martín-Gómez et al., 2015), although organic contamination on water isotopic measurements is still an unsolved problem (Wassenaar et al., 2018). Recently, new *in-situ* sampling techniques have been developed to analyse components of the water cycle that have eluded researchers until now. For example, new sampling probes have been developed to quasi-continuously extract water vapour from tree stems or soils for real-time isotope analysis (Volkmann et al., 2016b; Oerter and Bowen, 2017), to analyse gas exchange at the leaf level (Dubbert et al., 2014; Volkmann et al., 2016a) and to partition evapotranspiration in woody plants (Wang et al., 2010). These technical advances allow for continuous and unattended isotope measurements and hold promise for advancing our understanding of water storage dynamics, flow pathways and exchange processes in the critical zone.

## 3 Current knowledge, limitations and challenges

At the workshop, current knowledge and several key challenges in isotope-based studies of water flow pathways and plant-soil interactions in the critical zone were identified and discussed. Here, we summarize the known sources of isotopic variability in ecohydrological compartments and the related challenges in three main themes: methodological and conceptual limitations, heterogeneity in catchments and terrestrial ecosystems, and scaling issues.

## 3.1 Methodological and conceptual limitations: sampling the right water pool

Questions such as "What are the preferential water sources for plant root uptake?"; "To which extent do soil physical properties (e.g. soil texture, percentage pore space, soil organic matter content) influence the isotopic composition of water

taken up by plants?" and "What are the implications of different transport times and water storage within the rooting zone and within plants?" are central to understand water fluxes in the critical zone, and they have been explored in many stable isotope studies (e.g., White et al., 1985; Dawson and Ehleringer, 1991; Stahl et al., 2013; Bowling et al., 2017; Evaristo and McDonnell, 2017). While isotope measurements have become more accurate over the years and progress has been made in quantifying the proportions of different water sources by using Bayesian mixing models (such as SIAR, MixSIAR; see, for example, Evaristo et al., 2017; Rothfuss and Javaux, 2017), many conceptual and methodological challenges remain.

For example, some studies conducted at the catchment scale have found that water taken up by roots was isotopically different from stream water and groundwater (Brooks et al., 2010; McCutcheon et al., 2017). This suggests either that source-water sampling was incomplete, that fractionation processes modified the isotope composition of the water taken up by plants before or during uptake itself, or that other methodological issues may limit the utility of stable isotopes in tracing ecosystem water fluxes. Vital here is the issue of how to appropriately sample and extract water from soil and plant tissues. Several studies have shown that water from the same soil or xylem sample can have different isotopic compositions when extracted with different techniques. For example, soil water extracted with tension lysimeters may be isotopically different from that extracted with cryogenic distillation (Landon et al., 1999; Koeniger et al., 2011; Orlowski et al., 2016b; Gaj et al., 2017b; Thoma et al., 2018) and plant water extracted cryogenically may differ from water directly sampled from the xylem (e.g., Volkmann et al., 2016b; Zhao et al., 2016). Isotope ratios may also differ with different extraction times (West et al., 2006). It has been suggested that these differing signatures may represent different fractions of the total soil- or plant-water reservoir. What is lacking, but urgently needed (Berry et al., 2017), is to (i) develop well-tested and standardized sampling, extraction and isotope analysis protocols; and (ii) verify whether these extraction methods faithfully return the water pools we actually aim to analyse.

Another issue that needs to be addressed is the pore-scale variation in soil water isotopic composition. It has been suggested that differences in soil water isotopic composition depend on the soil water potential in different pore spaces, often referred to as "tightly bound" versus "mobile" water. Note that this language can be misleading because plants can use both. For example, plants sometimes rely on obtaining less-mobile matrix water that may have an isotopic composition distinct from the mobile gravity-drained water fraction that transits from the hillslope to groundwater and to streams (Brantley et al., 2017). However, soil and subsurface waters of different mobility were also found to be isotopically similar, and isotopic differences in these water sources can occur for reasons not related to mobility (McCutcheon et al., 2017). This suggests that our current perspective on why subsurface waters may vary isotopically is still limited. Currently, extraction of bulk water from soils or other subsurface compartments is done on relatively large soil sample volumes (50 cm$^3$ and above; Sprenger et al., 2015). However, experimental designs and methods to target the isotopic composition of water that is bound with different potentials and distributed in different pore sizes on scales below 50 cm$^3$ are needed to test whether such small-scale differences exist and to subsequently represent them in mixing models. If soil water isotopic composition cannot explain the

observed isotopic signature in plant (xylem) water, then other factors (e.g., plant physiological processes such as fractionation at the soil-root interface, uptake of dew, mist and/or fog via leaves and bark, evaporation through the bark, or mixing of xylem and phloem water) may explain these patterns (Eller et al., 2013; Berry et al., 2014; Martín-Gómez et al., 2016; Sprenger et al., 2017; Lehmann et al., 2017; Dawson and Goldsmith, 2018; Lehmann et al., 2018;).

Similar conceptual constraints relate to measuring the isotopic composition of water in plant tissue. It can take hours to days (or even months; Meinzer et al., 2006) for water absorbed by tree roots to reach the leaves (Dye et al., 1992; Ubierna et al., 2009). Further, water can be stored in the sapwood and outside the water transport pathway for days, particularly in conifers (Waring and Running 1978, Meinzer et al., 2006). Thus, the isotopic composition of xylem water may not always reflect the current water source(s) used by plants but may instead be influenced by water taken up days or even months beforehand (Brandes et al., 2007; Treydte et al., 2014). Recent experimental and modelling studies have revealed that xylem isotopic signatures also vary on short, sub-daily time scales (Volkmann et al., 2016a), but that water taken up by plants may be a mixture of both young water (from the current growing season) and old water (precipitation from the previous year), also depending on the time of the year (Brinkmann et al., 2018; Sprenger et al., 2018a). So far, most studies have assumed –not explicitly but often implicitly– some kind of steady-state conditions when trying to determine tree water uptake patterns. In addition, considerable spatial variation in xylem isotopic signatures within trees has been observed with values differing around and along tree stems (Cernusak et al., 2005; Volkmann et al., 2016a,b) and between stem and branch water (Dawson and Ehleringer 1993; Cernusak et al., 2005; Ellsworth and Williams, 2007; Zhao et al., 2016). How these temporal and spatial variations in plant isotopic signatures can inform end-member determination, and how they can be integrated into mixing models, remain unsolved issues at present.

The "two water worlds hypothesis" (McDonnell, 2014) has challenged the assumption of complete subsurface mixing that underlies many catchment models (Pfister and Kirchner, 2017). This hypothesis postulates that more mobile soil water contributes to groundwater recharge and streamflow whereas "tightly bound" water tends to be used by plants (McDonnell, 2014; Evaristo et al., 2015). Preliminary evidence from catchment studies based on the dual-isotope approach showed that bulk soil water was isotopically different from tension lysimeter water collected at the same depth (Brooks et al, 2010); that shallow soil water pool utilized by plants differed in isotopic composition from precipitation, stream baseflow, and soil-lysimeter water pools (Goldsmith et al., 2012); and that xylem water may be isotopically similar to soil and rain water, but different from streamflow and groundwater (Penna et al., 2013). The "two water worlds" hypothesis has stimulated new interpretations of ecohydrological data and new research questions to investigate water flow pathways in catchments (McDonnell, 2014) but also calls into question how often such dichotomous conditions exist in natural systems (Brantley et al. 2017). Currently, there seems to be a tendency to focus on just confirming or rejecting this one hypothesis although, as outlined in Berry et al. (2017) and Sprenger et al. (2016), alternative hypotheses need to be developed and tested to improve our current understanding. Because water held in the rooting zone or moving through that zone and other subsurface layers is

a continuum, where water transport is driven by gradients, and not separate "worlds", we see the necessity to move from the simplistic "two water worlds hypothesis" to an "*n* water worlds" concept, where multiple water reservoirs and flow pathways are invoked and parameterized, doing justice to the distribution of the different substrate types and sites. A challenge is that we are currently lacking easily applicable methods to sample the isotopic composition along this continuum. This can limit (and bias) sampling to only highly "mobile" and "bulk" soil water sampled by either tension lysimeters or cryogenic extraction and direct equilibration, respectively. Very recent findings highlighted that the relative contributions of mobile and less-mobile (bound at a wide range of water tensions) water are temporally variable and that the mobile water does not reflect the total plant-available water (Berry et al., 2017; Sprenger et al., 2018b). This should not be surprising: if water is highly mobile then it passes by roots too quickly for them to use. Although efforts have been made to compare different methods to sample water of different mobility for isotopic analysis (e.g., Geris et al., 2015; Orlowski et al., 2016b), we still lack clear definitions for distinguishing these multiple water pools. In addition to these methodological constraints, we also need to enhance our understanding of what pools of water plants might use and why. Some suggest that plants will use more "tightly bound" water even when more easily accessible, mobile water is available. As discussed by Bowling et al. (2017), plant water uptake and transport within the plant are primarily physical processes driven by a potential gradient between soil and leaf. Thus, this notion of plants using "tightly bound water" is inconsistent with the well-established mechanisms of water uptake and transport in plants i.e. via water potential gradients where plants are known to take up whatever water is most easily accessed (highest water potential) if they in fact have functional roots there (Dixon and Joly, 1896). The assumption that plants take up "tightly bound" water (as indicated by isotopic evidence) in the presence of less "tightly bound" water near the roots violates current physiological understanding about the mechanism of water uptake. Likely, other mechanisms, still not fully appreciated, affect the isotopic composition of water in plants. Therefore, we should inquire into them, instead of invoking a notion about plant water uptake that is inconsistent with previous investigations on plant physiology.

## 3.2 Heterogeneity in catchments and ecosystems

Given that the natural environment is heterogeneous on all scales, that sampling is by definition incomplete, and that the analytical process adds random errors (and, often, systematic biases), isotopic data are inherently subject to uncertainties. While many of our research questions are specifically focussed on exploring heterogeneous patterns across different domains (e.g., different climatological conditions, soil types or vegetation types), there is also considerable variability within each domain. Observed differences in the isotopic composition of water in various compartments of a catchment or an ecosystem are the result of many, often simultaneous, processes. The isotopic compositions of water samples are often shaped by mixing as well as fractionation processes; both mixing and fractionation can occur in different compartments (e.g., soils, plants, atmosphere) either simultaneously or at different times as water passes through the system. Because we still lack a thorough understanding of the underlying mixing and fractionation processes and of the spatial and temporal scales at which they operate, difficulties remain in interpreting the isotopic compositions that we measure in our environmental samples.

While early isotope applications in catchment studies demonstrated the importance of considering temporal variability in precipitation and runoff (Kendall and McDonnell, 1993), less attention was initially paid to spatial heterogeneity, especially at the hillslope or small catchment scale, under the assumption that streamflow inherently integrates over spatial variations in the upslope contributing area. However, it is now well known that characterising the spatial patterns in isotopic composition (so-called "isoscapes") is another important tool towards increasing our understanding of hydrological processes (West et al. 2009; Bowen and Good, 2015). In addition, applications in ecohydrology often require spatially explicit characterizations of soil water, groundwater, and plant water, which do not necessarily integrate across large spatial areas and thus reflect local heterogeneity. In the following, we highlight the main sources of heterogeneity that characterize different water pools relevant to ecohydrological studies.

Precipitation represents a major source of spatio-temporal heterogeneity that results in variations in all subsequent biological and hydrological compartments. The combined effects of variability in atmospheric parameters such as humidity, temperature and solar radiation influence the isotopic composition of precipitation, manifesting in temporal (Dansgaard, 1964; Rozanski et al., 2013; Coplen et al., 2008; Coplen et al., 2015; Munksgaard et al., 2012) and spatial (Ingraham, 1998; Bowen & Revenaugh, 2003; Bowen, 2008; Fischer et al., 2017; Allen et al., 2018) variability at multiple scales. However, at least at the plot scale (i.e., tens of meters) and in the absence of significant altitude variations, the spatial variability of precipitation is usually of minor importance. Precipitation is often collected by tipping buckets with a ~200 mm diameter (Fig. 1), which is assumed to integrate small-scale variations and be representative for the plot scale. At larger scales (e.g. > hundreds of meters), we might observe systematic variations that are functionally relevant, and obtaining the isotopic composition of an input that is representative for the entire study area can be challenging. Temporal variability can also be very pronounced, even during a single storm event, and quasi-continuous precipitation sampling is vital for capturing such a variability in the input signal to the system (Munksgaard et al., 2012; Pangle et al., 2013; von Freyberg et al., 2017).

Canopy interception of liquid water involves flow through a small storage with short mean residence time, largely resulting in throughfall and stemflow having patterns of temporal variability that resemble the initial precipitation inputs (e.g., Ikawa et al., 2011). Nevertheless, storage and subsequent evaporation of intercepted precipitation can result in below-canopy inputs to soil that differ from open precipitation by more than 2‰ in $\delta^{18}O$ for single events and over longer periods (Allen et al., 2017). There can also be spatial variations of several ‰ in $\delta^{18}O$ because stemflow and throughfall dripping points involve long residence times (Allen et al., 2014) and thus have distinct isotopic compositions which are challenging to capture.

In places where snowfall is an important component of precipitation, snow accumulation dynamics can significantly modify the spatio-temporal patterns of precipitation isotopic composition. Snowpack depth and density are known to be very irregular, following complex compaction and redistribution dynamics that are influenced by topography, wind and vegetation (e.g., Trujillo et al., 2009). As a consequence, snowmelt is very heterogeneous and its flow pathways change

through time as the snowpack evolves. Due to these dynamics, the spatial variability in the isotopic composition of snowpacks and snowmelt can be very large (Rücker et al., 2019; Webb et al., 2018). Moreover, melt-and-refreeze dynamics during water percolation through the snowpack cause heterogeneous and time-variable isotopic fractionation (Taylor et al., 2001). Canopy-intercepted snow can have longer residence time than liquid water and, because fractionation due to sublimation and refreezing is greater, especially at lower temperatures, it may contribute to larger isotopic changes (Koeniger et al., 2008). All these processes often interact and make the representative characterization of snow and especially snowmelt isotopic composition highly difficult.

Water flow and transport through heterogeneous porous media are complex processes that still represent a hot topic for the vadose zone and groundwater communities (Kitanidis, 2015). However, not only is subsurface flow always heterogeneous (Gehrels et al., 1998; McDonnell et al., 2007; Stumpp et al., 2007; Troch et al., 2009; Stumpp & Maloszewski, 2010), but also fractionation processes in the subsurface vary in both time and space. Evaporation, which is largely controlled by surface energy variations, is a major contributor to isotopic fractionation, especially at shallow soil depths. Generally, evaporative effects decrease with increasing soil depth, resulting in deuterium excess or line-conditioned excess of soil water becoming less negative with depth (e.g., Sprenger et al., 2017). Moreover, bulk soil water shows more evaporative effects than lysimeter water and may be characterized by values of line-conditioned excess below zero (McCutcheon et al., 2017), because more "tightly bound" waters integrate older ages and, therefore, are affected by kinetic fractionation during periods of atmospheric evaporative demand (Sprenger et al., 2017). However, bulk soil water, extracted by cryogenic distillation or direct equilibration (Sprenger et al., 2018c), which contains both mobile and matric-bound water, is generally more depleted in heavy isotopes than mobile water collected by tension lysimeters at the same depth and location, although it is unclear how much, volumetrically, the "bound water" fraction in the bulk sample is, what isotopic impact it has on the final measured isotope ratio, and how much of it a plant might or can use (Brooks et al., 2010). Moreover, deep bulk water is usually more depleted than mobile water during spring or summer, due to filling of fine pores of a relatively dry soil with depleted precipitation several months earlier (Geris et al., 2015; Oerter and Bowen, 2017; Sprenger et al., 2017). This suggests that old and more "tightly bound" water might show not only a distinct isotopic signal compared to mobile water due to seasonally variable precipitation inputs, but also an evaporative enrichment signal from periods of high evaporative demand (Sprenger et al., 2017).

In soils and groundwater, isotopic heterogeneity results from differences in the inputs (precipitation, throughfall, snowmelt), differences in the temporal integration of previous precipitation events (Yang et al., 2016), and differences in the subsequent fractionation from evaporation and transpiration (Benettin et al., 2018). As a general rule, smaller storage-to-output ratios with short residence times generally lead to higher temporal variability. Conversely, larger storages with longer residence times are likely to lead to a more dampened signal that integrates over longer periods of time (e.g., Zhang et al., 2016; Benettin et al., 2017). Obtaining representative samples in soils is challenging also because soil water content, soil texture,

mineral composition, and the content of organic matter are spatially heterogeneous and strongly influence how soils interact with water molecules (e.g., Barnes and Allison, 1983; Oerter et al., 2014; Oshun et al., 2016; Gaj et al., 2017a). Hence, the interaction with soil particle surfaces (Lin et al., 2018), soil organic matter (Orlowski et al., 2016a), local soil properties (Yang et al., 2016), microorganisms (Blake et al., 1997; Kool et al., 2007) and plants (Vargas et al., 2017) may introduce additional isotopic heterogeneity. Also the origin of soil water is diverse, comprising a mixture of precipitation events from different times, sources (air masses), and types (e.g., rainfall, snow, hail); it may also include groundwater and, in the case of agricultural fields, irrigation water derived from groundwater, lakes, or rivers. With increasing soil depth and down to groundwater, we generally expect that dispersive transport will lead to increasingly damped spatio-temporal variations around the average input composition. Nevertheless, we commonly characterize groundwater with wells that receive water from a variety of depths that may have isotopically distinct waters (Jasechko et al., 2017). While it is generally assumed that groundwater integrates inputs over time and space (Scheliga et al., 2017), the integrations may be short and small when one distinguishes shallow groundwater or perched water tables from soil water (Uhlenbrook and Hoeg, 2003). Smaller-scale spatial variations in groundwater isotopic composition are typically not well characterized, and ecohydrological applications often assume spatial homogeneity.

Plant xylem water reflects the spatial and temporal heterogeneity of the water that is accessed by functional plant roots (Ehleringer and Dawson, 1992). Due to differences in functional rooting locations among species and individuals, plant water isotopic composition is often distinctly different and highly variable among different species (Bertrand et al., 2014; Schwendenmann et al., 2015; Volkmann et al., 2016a). However, heterogeneity in soils and other subsurface compartments with respect to texture, structure, and water content could also result in differences in xylem water isotopic composition across trees with similar rooting patterns. Within-plant variations in xylem water composition may also occur because travel times can increase with within-plant path length, so temporal variations in source isotopic composition must result in vertical variations in xylem isotopic signatures (Dawson and Ehleringer, 1993; Cernusak et al., 2005; Zhao et al., 2016). Lateral or radial variations in the trunk occur due to radial variations in the source water composition or potentially in water transport rates (and thus water age), depending on the degree of sectoriality of a plant's xylem transport (Steppe et al., 2015; Volkmann et al., 2016b). Even among individual vessels, variations are expected, given the little evidence of dispersion across xylem flow pathways provided by different studies (Zimmerman and Brown, 1971; Kline et al., 1976). Once water in the xylem reaches the stomatal aperture of the leaves, the vapour pressure deficit between the ambient air and the intercellular cavities results in isotopic fractionation that significantly alters the source signal (Dongmann et al., 1974). These fractionation effects are spatio-temporally variable (see, for instance, Helliker and Ehleringer, 2002), but may be masked by wood and other tissues that might act as temporal and spatial integrators of heterogeneous processes in leaves (Gessler et al., 2014; Singer et al., 2014).

**3.3 Scaling issues**

Many of the issues raised in the above sections relate to small-scale processes of water flow and transport, as research to date has often been performed at the level of soil patches or individual plants. Less attention has been directed towards determining how small-scale ecohydrological processes can be used to understand catchment- or landscape-scale phenomena. For example, Bertrand et al. (2014) found that trees used different water sources depending on their location within an alluvial system, whereas Gaines et al. (2016) did not detect differences in root length density and water uptake

among trees located at different hillslope positions. Hsueh et al. (2016) showed that trees on deltaic hummocks preferentially took up water from unsaturated hummock tops to protect from higher salinity and saturated soil in swales and the lower portions of hummocks. Dudley et al. (2018) saw little evidence to suggest that landscape position altered groundwater uptake by shrubs. Pettit and Froend (2018) showed that riparian trees located on relatively shallow groundwater had greater growth rates, larger diel responses in stem diameter and were less reactive to extended dry periods, than trees located in areas of

deep groundwater. Sprenger et al. (2018c) found that soil water beneath conifer trees was more fractionated than beneath heather shrubs or red oak trees, and that sampling locations closer to streams had a more depleted isotopic composition than hillslope sites, revealing increased subsurface mixing towards the saturated zone and a preferential recharge of winter precipitation. Such differences in the depths and types of water taken up by trees may have a critical impact on streamflow and hydrochemistry (Brantley et al., 2017), both of which generally depend on travel times and flow pathways (Rinaldo et

al., 2015; van der Velde, 2015). Studies that systematically monitor ecohydrological processes across environmental conditions, soils, and vegetation types within landscapes will certainly be needed in the future, but a challenging question today is how representative such individual tree/plant studies are for larger-scale systems. Answering this question will require us to understand both the heterogeneity in ecohydrological processes and the resulting heterogeneity in the isotopic composition of water when moving up in scale. As we move from, e.g., the individual to stand or hillslope scale, systematic

sampling approaches that account for this heterogeneity within a landscape element of interest will have to be tested. For example, these approaches might identify a sample size that is statistically representative and integrates over the main sources of variation within a given system.

If such an approach exists for a given process of interest, it could inform sampling protocols. As we move further up in scale

of, e.g., catchments or an entire landscape, we need to develop approaches that appropriately represent this heterogeneity in hydrological and ecological models. Here, much may be learned from work that has been carried out in soil science (Lark, 2012a, b) on how to investigate spatial patterns and scaling related to isotopic studies in ecohydrological systems.

Furthermore, a gap exists between the scale at which we typically apply our observational isotope techniques and the range

of spatial and temporal scales across which we draw ecohydrological inferences. Repeated observations have provided insights into the heterogeneity of hydrological and ecological processes at these larger scales, but they have also revealed the

limitations in our current sampling strategies. In defining the isotopic signal of the measured ecohydrological compartments as their variations across spatial and temporal scales, and the noise as the variance of repeated sampling of the same water pool, the key issue is to determine which sampling strategies can ensure the best signal-to-noise ratio, i.e., that allow for the signal to be many times greater than the noise. However, inevitably, our ability to observe the variability within a sampling event is a product of the duration and size of that sampling. In practice, we often rely on a few samples to characterize a much larger heterogeneous domain. For example, small samples of xylem tissue are commonly used to characterize an entire forest or a few soil cores are meant to represent the entire range of spatial variation in soils. Temporally variable processes are often sampled in a few short measurement campaigns, yet they are assumed to be representative of much longer (and perhaps variable) periods. Furthermore, investigators should consider whether concurrently sampled components of the hydrologic cycle are representative of the same time (e.g. leaf water may be sourced from soil water taken up by roots days or weeks prior to the day of sampling). Inadequately sampling heterogeneous can lead to interpretation errors. This problem is exacerbated when analysis methods do not properly reflect the statistical properties or uncertainty of the sample pool. For example, single mean values are often used as end-members in mixing models, which exaggerates the precision of the source partitioning (Phillips and Gregg, 2001). We recommend that researchers adopt sampling strategies to determine the signal variances and to deliberately integrate across the smaller-scale variability so that heterogeneity across the ecohydrological domains of interest can be appropriately characterised. This includes, for instance, planning sampling campaigns that span over multiple (growing) seasons or that comprise multiple locations within the same landscape element (e.g., different hillslope positions), finding the correct balance between the necessary labour and cost, and the additional information provided. For example, this would allow for considering mean values with their variability as end-members in mixing models, thus more appropriately addressing the uncertainties arising from the intrinsic spatio-temporal variability in the studied system.

## 4 New perspectives and research opportunities

The current constraints and knowledge gaps we have presented above can also be seen as opportunities for new ecohydrological research. In this section, we outline future directions for research into water fluxes and partitioning using stable isotopes.

First of all – and perhaps most urgently – we call for systematic comparisons and methodological reviews of techniques for extracting water from xylem, soil and other rooting media in the subsurface, followed by the development of standard protocols. Recent experimental work has attempted to evaluate the cryogenic vacuum distillation method for soil and plant water extraction and critically discussed its suitability as a standard method for plant-water investigations (Orlowski et al., 2013 and 2018a, b; Newberry et al., 2017; Thoma et al., 2018). Some of these studies have shown that the extraction method can have a significant effect on the isotopic value obtained from the analysis of pore water, depending also on the soil type

and organic matter content (Sprenger et al., 2015; Newberry et al., 2017). However, the protocols are not always applied in comparable ways and some methods such as drying soils at excessive (105°C) temperatures will introduce artefacts because material properties change when the strongly adsorbed water is removed, which would not happen in field soils that plants are actually rooted into. It has been suggested that future work should examine how the full range of cryogenic extraction conditions (extraction time, temperature, vacuum threshold) as well as physicochemical soil properties affect the isotopic composition of extracted water (Orlowski et al., 2016 a, b; 2018; Gaj et al., 2017b). For instance, recent results (e.g., Gaj et al., 2017a) suggest that water from different soil types should be extracted with different temperatures to extract the same water pool, but investigations on the range of needed temperatures for each soil type are still needed. Comprehensive inter-comparisons of soil water sampling and extraction methods (including, for instance, techniques such as tension lysimeters, high pressure mechanical squeezing, centrifugation, direct vapour equilibration, microwave extraction, and cryogenic vacuum distillation) and xylem water extraction methods (such as wood cores, pressure vacuum, centrifugation, Scholander-type pressure chambers as well as direct vapour equilibration and cryogenic vacuum distillation) are urgently required to develop standardised sampling and extraction protocols (Millar et al., 2018).

More specifically, we believe it is critical to set up experiments that will allow us to understand whether the observed differences in isotopic composition of extracted waters reflect isotopic variations in the real world or are instead associated with sampling and/or analytical artefacts. We need to work towards a better understanding of how to extract the particular water from soils and plant tissue that is relevant to answer our specific research questions. How do we extract the soil water that takes part in water flow processes? From which plant tissue should we extract water? This calls for a more detailed analysis of which water pool each method is able to access, because different methods can sample different waters in both soils and plant tissues, therefore leading to potential differences in isotopic composition. For instance, cryogenic vacuum distillation can extract nearly all water from soil samples, even water held at tensions so high that plants cannot access it (although the volumes of this very "tightly bound" water are likely to be very small relative to the volumes plants use as mentioned above). In contrast, tension lysimeters typically sample water held at <200 kPa (Geris et al., 2015), and thus do not collect all the water that plants can access and take up (Sprenger et al., 2018b). Moreover, the differences in isotopic composition observed in soil water samples through tension lysimeters and all the water accessible to plants are time variant and are linked to the volume and age of the mobile water (Sprenger et al., 2018b). For plant samples, cryogenic vacuum distillation normally extracts all water from plant tissue, including intra-cellular water that is not part of the advective flow system. In contrast, other techniques (e.g., Scholander-type pressure chamber, vapour equilibration) are able to extract water from xylem vessels only (Volkmann et al., 2016b). In addition to the above-mentioned aspects, the extraction and analytical methods used (e.g., extraction technique, temperature, time, number of replicates, laser or mass spectrometer used) need to be thoroughly documented and reported.

Secondly, we call for more high-resolution monitoring and extensive labelling experiments with known boundary conditions (e.g., Kulmatiski et al., 2010; Grossiord et al. 2014; Beyer et al., 2016; Priyadarshini et al., 2016). These would facilitate more rigorous observations of physiological and ecohydrological processes and a more detailed characterization of the spatial heterogeneity and temporal dynamics of isotopic composition in different compartments of the critical zone. As an example, sampling xylem at high temporal frequency at specific stem positions might provide more detailed information about fractionation processes during water transport from root to shoots. At the same time, limitations that might not be possible to overcome using natural isotopic abundances (e.g., the differentiation of isotopically similar water sources) can be addressed using isotopic labelling (Koeniger et al., 2010). The usefulness of labelling studies has been acknowledged for decades, also coupled to modelling approaches (e.g., Stahl et al., 2013), but the combination of high-resolution monitoring with labelling leads to a new dimension of research opportunities. Indeed, labelling and high-resolution monitoring experiments have the potential to provide new insight into the size and speed of water flow pathways in both soils and plants. While high-frequency measurements of isotopes in soil water have been often reported, *in-situ* measurements of xylem water isotopes remain challenging (Martín-Gómez et al., 2015; Volkmann et al., 2016b). Resolving this limitation would be a major step towards broadening the range of time scales that can be investigated. This also requires a thorough examination of how organic compounds in plant waters may distort laser spectroscopy measurements of isotopes in water (West et al., 2010; 2011). There is also great potential for studies using two or more tracers simultaneously ("dual-labelling"). For instance, different soil layers might be labelled with different tracers (e.g., "high" deuterium label on the surface, "low" oxygen label at depth) to explore which water pools plants preferentially access under variable conditions (see, for example, Bachmann et al., 2015). In this regard, research on the often-raised issue of water vs. nutrient availability could improve our current understanding of ecohydrological feedbacks (Bakhshandeh et al., 2016).

The third main aspect highlighted during the workshop's discussions is the need to incorporate knowledge regarding fractionation effects (e.g., Dawson and Ehleringer, 1993) into the models that are used to interpret isotope data. For example, process-based models may help interpreting observations and experimental data (Benettin et al., 2018), assessing the importance, seasonality and uncertainty in evapotranspiration partitioning (Knighton et al., 2017; Smith et al., 2018), and characterising water pathways and quantifying the associated travel times at the catchment scale (Kuppel et al., 2018). Clarifications are needed on which parameters to include in a model and on when it might be possible to ignore their influence. This knowledge is still lacking, which may lead to incorrect interpretations of data and development of unnecessarily complex models. This knowledge will also produce better estimates of the uncertainties associated with isotope data and better methods to propagate them. Uncertainties, also related to fractionation effects, should also be applied in Bayesian mixing models, which are used to quantify the proportional contributions of various sources to a mixture (Davis et al., 2015; Evaristo et al., 2017). For instance, Rothfuss and Javaux (2017) examined the uncertainty associated with different types of mixing models, stating that graphical and statistical methods have major drawbacks when analysing root water uptake depths. They found that the latest generation of Bayesian mixing models performs well for that purpose, but

only when the number of considered water sources in the soil is high and closely reflects the vertical distribution of the soil water isotopic composition. Additional tracers can be helpful to support and strengthen the observations obtained by using stable isotopes. Trace elements taken up through plants (e.g., gold particles, Lintern et al., 2013) might have particular potential for inferring root water uptake. Chemical tracers (e.g., Haase et al., 1996) and tritium (Zhang et al., 2017) have also been used to study water uptake depths. However, using other types of tracers, such as fluorobenzoic acids, dissolved ions, and isotopic ratios of other elements such as radium or strontium, will introduce further complexity to the system due to potential interactions of these tracers with soil, roots and the water itself. Such effects need to be carefully studied in order to provide meaningful interpretations. Even the use of labelled water can produce artefacts, for example masking fractionation processes that in turn can influence the results. Therefore, there is a need to understand the conditions that limit the use of stable isotopes as tracers in ecohydrological applications and to pinpoint the processes for which they may not be the best tracers. By carefully matching the methods with the research objectives, we can assess the reliability of stable isotopes of hydrogen and oxygen and determine whether integrating isotopes with additional tracers would be helpful.

We strongly recommend designing studies that are not overly sensitive to the intrinsic uncertainty of the domain of interest and that represent heterogeneity in a way that costs (i.e., labour) and benefits are balanced. Potential solutions include the use of highly controlled settings, using tracer injections to amplify the signal, constraining the spatial or temporal domain of a study, determining whether end-members are sufficiently (and consistently) distinguishable, asking coarser questions, or simply anticipating the higher costs associated with collecting more samples than are conventionally used. While we often do not quantify variations within a sampled water pool, the uncertainties associated with the (hypothetical) effects of within-sample variations should also be considered more consistently in analyses and interpretations. This includes, for instance, making efforts to better quantify the variance of the isotopic signal within and between water pools by increasing (in time and/or space, depending on specific research questions) the number of collected samples.

Finally, the ubiquitous presence of hydrogen and oxygen isotopes in different compartments of the critical zone (atmospheric water, subsurface and surface waters, plant tissues) and the close linkages between physical processes in the biosphere, lithosphere, atmosphere and hydrosphere inherently call for new interdisciplinary isotope-based investigations. Posing research questions from an interdisciplinary perspective can help to achieve a more comprehensive interpretation of data and results, and a more detailed understanding of the physical processes involved. In this context, interdisciplinary research can help us to understand in detail the conundrum provided by isotopic evidence that suggests that at least some plants access "tightly bound" water more easily than the "mobile water", violating well-established physiological knowledge, and to stimulate research questions about the mechanisms leading to the observed isotopic values in subsurface waters and xylem. We encourage collaborations among ecologists, plant physiologists, hydrologists, hydrogeologists and soil scientists to achieve a broader perspective from different points of view on water fluxes in the critical zone. We particularly advocate for

new interdisciplinary studies into controls on spatial and temporal patterns of ecohydrological fluxes for different plant species, in different landscapes, and under different climatic forcing.

## 5 Concluding remarks

The workshop on "Isotope-based studies of water partitioning and plant-soil interactions in forested and agricultural environments", held in Italy in September 2017, offered scientists with different backgrounds the opportunity to meet and share ideas, experiences, and perspectives on studies of water fluxes in the critical zone based on stable isotopes of hydrogen and oxygen. The past decade has seen the emergence of new instruments and new insights, oftentimes questioning the simplifications we were forced to make earlier, but at the same time opening our eyes to new and important sources of variation. Although the need to re-evaluate our methods was a consistent theme, the opportunities provided by continuous measurements are very promising. Within the workshop and the scope of this paper, our effort has been to convert these identified knowledge gaps into new interdisciplinary research opportunities that can pave the way towards a better understanding of the physical processes governing water movement in natural and anthropogenic terrestrial environments. We believe that interdisciplinary discussion of these themes is useful for the entire ecohydrological community to foster collaborations and to develop suitable methods to take full advantage of the stable isotopes of hydrogen and oxygen as an effective tool to investigate the fate, availability and the distribution of water in the environment.

## Author contribution

DP, LH, and FS organized the workshop, wrote the outline and the first draft of the manuscript, supervised the whole writing process by integrating the corrections and comments of the other authors and critically revised each draft version. STA, PB, MB, JG, JWK, JDM, LS, THMV and JvF wrote specific sections of the manuscript. AA, NC, ME, JF, YG, JJM and GZ provided further specific comments and literature references. PL, RTWS, TED, and JWK led the discussion groups during the workshop, edited the text adding relevant specific comments, contributed to reach uniformity and coherence during the whole process, and improved the language stylistically and grammatically.

## Competing interests

The authors declare that no competing interests are present

## Acknowledgements

The authors thank Marialaura Bancheri, Michele Bottazzi, Roman Cibulka, Massimo Esposito, Alba Gallo, Cesar D. Jimenez-Rodriguez, Angelika Kuebert, Ruth Magh, Stefania Mambelli, Alessia Nannoni, Paolo Nasta, Vladimir Rosko,

Andrea Rücker, Noelia Saavedra Berlanga, Martin Šanda, and Anna Scaini for their contributions during the discussion at the workshop "Isotope-based studies of water partitioning and plant-soil interactions in forested and agricultural environments". The authors also thank "Villa Montepaldi" and the University of Florence for the access to the workshop location, and the municipality of San Casciano in Val di Pesa for logistical support. The authors thank the Department of Innovation, Research and University of the Autonomous Province of Bozen/Bolzano for covering the Open Access publication costs. Last, but not least, the authors wish to thank Matthias Sprenger, Stephen Good and J. Renée Brooks, as well as the Editor David R. Bowling, whose constructive reviews greatly improved this manuscript.

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

**Figures**

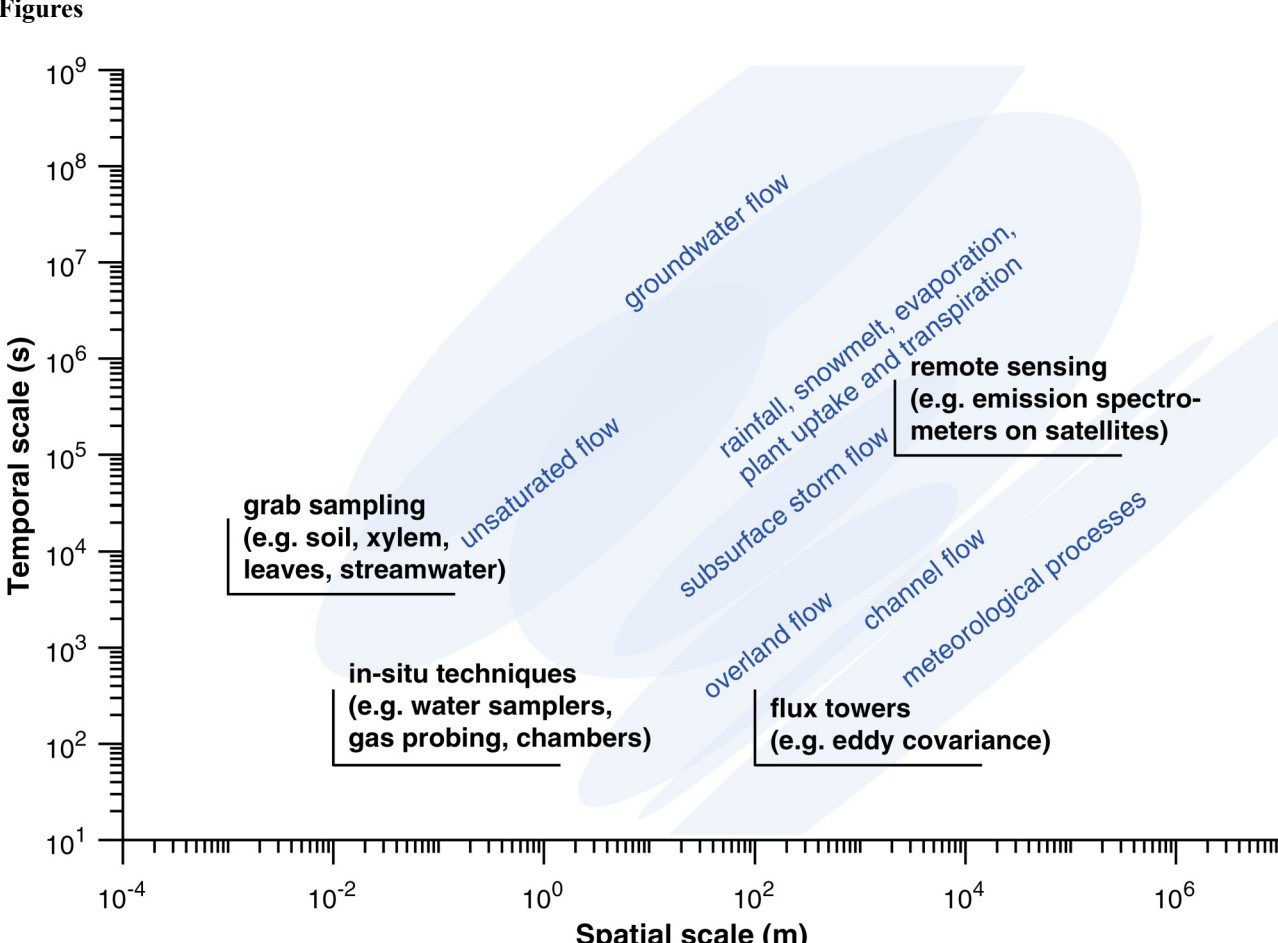

**Figure 1:** Simplified representation of spatial and temporal scales of ecohydrological processes (based on Blöschl and Sivapalan, 1995 and Bowen and Good, 2015) and of isotope observational techniques frequently used to characterize these processes. Method scales represent typical minima that may be expanded through multiple observations, and process scales approximate characteristic scales of variation. While recent developments in sample acquisition and analytical techniques have increased sample throughput and pushed the limits of observational capacity, a lack of ability to immediately characterize heterogeneous hydrological and ecological processes at typical study scales is glaringly apparent. As a consequence, large uncertainty and interpretation errors can result in isotopic studies, and open questions exist regarding appropriate sampling strategies.