# Peer review of "Ideas and perspectives: Tracing terrestrial ecosystem water fluxes using hydrogen and oxygen stable isotopes – challenges and opportunities from an interdisciplinary perspective"

_Biogeosciences, 2018_

## Referee Comment (RC1) · M. Sprenger (Referee) · 10 Jul 2018

I am happy to read that Daniele Penna and others attending the workshop "Isotope-based studies of water partitioning and plant-soil interactions in forested and agricultural environments" share the results of their discussions in the submitted commentary ("Ideas and perspectives" as called in Biogeosciences). They provide an adequate overview of the current state regarding recent developments in the application of stable isotopes as a tracer in biogeosciences and discuss also limitations that we are facing at the moment. I like how they frame these limitations as new perspectives and research opportunities to put a positive spin on these challenges.

I also welcome their aim to promote interdisciplinary research using stable isotopes. The commentary is in the scope of the journal and I am sure it will be of interest to a broad community using stable isotopes. I suggest publication after minor revision and I provide some critical comments below. Please note that the provided references are of course only meant as suggestions and that reference to my own work are mainly provided to underline my arguments given in the general comments.

**General comments**
I think that the commentary could be stronger if the authors would revisit some parts, as the structure is not very consistent. For example, some parts in Section 2 would fit better to Section 3, as they deal with limitations, while several paragraphs of Section 3 read like a review and would fit better in Section 2. I provide example of these paragraphs in the specific comments.
I would also encourage the authors to acknowledge recent developments that deal with their suggested research agenda, when asking for isotopic variability of groundwater (as done by Scheliga et al., 2017), promoting tests of isotope analysis for vegetation samples (as done by Millar et al., 2018), analysis of spatial variability of soil water stable isotopes (as done by Yang et al., 2016), suggesting dual-labeling studies (as done by Bachmann et al., 2015), incorporating evaporation fractionation in catchment models (as done by Knighton et al., 2017, Smith et al., 2018, and Kuppel et al., 2018). While these studies were partly published recently, I still think it would be worth looking into them and considering including them. This would provide examples of developments that might go into the direction that you are suggesting and help the reader to see what is currently being done and tested.
One comment on the "n water world": The "Two Water Worlds" were suggested based on the findings by Brooks et al. (2010) that soil waters extracted with suction lysimeter or cryogenically have different isotopic compositions. Thus, this definition of a split into two subsurface pools mainly stems from the limitation to not be able to extract water held at different pressure heads. I obviously agree that the subsurface is a continuum

of varying pressure heads (P6 L23). However, we are currently lacking the means to sample the isotopic composition along this continuum (water retention curve), but efforts have been made to compare different methods sampling waters of different mobility with different methods (see Figure 4 by Geris et al., 2015 and the work by Orlowski et al. (2016)). The relevance of different pressure heads for the TWW has been discussed earlier by Berry et al. (2017). The point that I want to raise here: Of course, it would be neat to be able to sample a "n water world", but is it really practical? Instead, the TWW is relatively loosely defined into "mobile" and "less mobile or tightly bound" waters. However, from a stable isotope perspective (on which the TWW is based), we can only distinguish between "mobile" and "bulk" soil water given our limitations in the sampling procedure with either suction lysimeter and cryogenic extraction/direct equilibration, respectively. I tried to convey this message with my co-authors in Sprenger et al. (2018b). Important with this regard is that i.) the relative contributions of "mobile" and "more tightly bound water" is temporally variable (Figure 5 in Sprenger et al., 2018b) and that ii.) the "mobile" water does not reflect at all the total plant available water, but plants can access more tightly bound waters than the suction cup lysimeter (Figure 1 in Berry et al., 2017 and Figure 1 in Sprenger et al., 2018b). There is further a lack of clear definition distinguishing the two water pools.

**Specific comments**
P2 L11: I am not sure if a report (2030 WRG, 2009) sponsored by "The Barilla Group, The Coca-Cola Company, The International Finance Corporation, McKinsey Company, Nestlé S.A., New Holland Agriculture, SABMiller plc, Standard Chartered Bank, and Syngenta AG" is an appropriate reference.
P2 L 29: To my understanding, it is not only that the laser-based instruments are more affordable, but also running it for analysis is cheaper and easier.
P3 L2: I think that the context is missing here. One might ask "which well-mixed conditions"? While I believe to know what you refer to, you aim to reach a wider audience in Biogeosciences and thus, should be clear about the simplifying assumptions.

P3 L25-L29: This reads more like "limitations and challenges" and is an example for what I mean that the structure could be improved.

P4 L17-L19: Again, when you start a sentence with "One limitation...", this might fit better in section 3 of the manuscript.

P4 L19: What about carrier gas issues like CO2, which potentially cause issues for CRDS (Gralher et al., 2018)? Beside consequences for the direct-vapor equilibration method, this would also be relevant for in-situ measurements via the vapor phase.

P4 L23: You mention soils, tree stems and leaf as examples. What about ET-partitioning as done for example by Wang et al. (2010)?

P5 L6: I think that the comparison and review study by Rothfuss and Javaux (2017) is a more appropriate reference here.

P5 L10: Given the methodological issues with the calculations of source waters by Evaristo et al. (2015), as revealed by Javaux et al. (2016), this might not be a good reference.

P5 L18: Consider including Zhao et al. (2016) as reference, who compared also different methods for xylem sampling.

P5 L20: Be more specific: What do you suggest to the community? What do you "aim to analyze" (- depends on your research question, or not)? How is this message different to the section 4.3 in the commentary by Berry et al. (2017)?

P5 L26: I am not aware that McCutcheon et al. (2016) studied the stable isotopic compositions among different pore spaces. Instead, they specifically stated that "We are not able to determine if pore-scale variability can impact isotopic composition of root-absorbed and draining water".

P5 L31: How about evaporation through the bark as studied by Martín-Gómez et al. (2016)?

P6 L7: Do you think numerical simulations can help assessing such ages, as done by Brinkmann et al. (2018) and Sprenger et al. (2018a)?

P6 L14: Do you think relating tree ring to source waters would be beneficial to study long term dynamics (e.g., Singer et al., 2014)?

P6 L25: See general comments regarding the "n Water World".

P7 L18- P8 L23: These paragraphs read like a review and do not link to "Limitations and challenges".

P8 L26: Please note the intensive sampling of groundwater stable isotopes by Scheliga et al. (2017).

P8 L29- P9 L10: To me, you miss pointing out the limitations and challenges in these paragraphs.

P9 L21: Regarding the heterogeneity of soil water stable isotopes, please see the data set by Yang et al. (2016) the intense variability across relatively small area.

P10 L23: Consider including the interesting study by Millar et al. (2018).

P10 L14: This ratio between water accessible for plants and water extractable by suction lysimeter is time variant (Sprenger et al., 2018b) – this is usually ignored and cannot be accounted for when sampling twice soil and plant isotopes over a year.

P12 L5: I suggest providing examples, where this has been done (e.g., Knighton et al., 2017, Smith et al., 2018, and Kuppel et al., 2018).

P12 L1: As done by Bachmann et al. (2015).

P12 L12: as reviewed by Rothfuss and Javaux (2017)

P12 L14: Can you provide any studies backing this idea? Is to "introduce further complexity" something positive in this context? This "potential interactions of these tracers with soil, roots and the water itself" would need to be carefully studied and understood for the tracer interpretations, right?

Fig. 1: You've added remote sensing to the graph, but do not discuss it in the text.

Fig. 1: Plant uptake and transpiration have scales > 10m in your graph, but much of your discussion deals with processes way smaller where water is taken up by roots (scales « 1m).

Fig. 1: How about potential of tree ring samples (in the category "grab sampling")?

Fig. 2A: It seems from Fig. 2B that the green shading represents the range of samples and the black line represents an average value. Are these values then representing average and range from replicates (e.g., five soil samples at 0-5 cm soil depth)?

Otherwise, how could you have a range from one sample value (as indicated on the y-axis)? Or does the green shade the range of values out there in nature and the black line represents the individual sample? This is not clear to me.

Fig. 2C: Not sure if I understand how the temporal variability of precipitation and throughfall is higher within events than among events. Is not the variability within the events part of the overall variability among events? Looking at Figure 6 in Freyberg et al. (2017) it seems that variability between events is higher.

Fig. 2C: Why did you not include plant water isotopes?

**Technical corrections**

Title: As I understand, the manuscript title must start with "Ideas and perspectives:" (https://www.biogeosciences.net/about/manuscript_types.html)

P2 L24: Repetition of "Stable isotopes of hydrogen and oxygen"

P14 L1: Coauthors missing for Bertrand et al. (2014).

**References**

Bachmann, D., Gockele, A., Ravenek, J. M., Roscher, C., Strecker, T., Weigelt, A., Buchmann, N., and Rixen, C.: No Evidence of Complementary Water Use along a Plant Species Richness Gradient in Temperate Experimental Grasslands, PLoS ONE, 10, e0116367, doi:10.1371/journal.pone.0116367, 2015.

Berry, Z. C., Evaristo, J., Moore, G., Poca, M., Steppe, K., Verrot, L., Asbjornsen, H., Borma, L. S., Bretfeld, M., Hervé-Fernández, P., Seyfried, M., Schwendenmann, L., Sinacore, K., Wispelaere, L. de, and McDonnell, J.: The two water worlds hypothesis: Addressing multiple working hypotheses and proposing a way forward, Ecohydrol., 11, e1843, doi:10.1002/eco.1843, 2017.

Bertrand, G., Masini, J., Goldscheider, N., Meeks, J., Lavastre, V., Celle-Jeanton, H., Gobat, J.-M., and Hunkeler, D.: Determination of spatiotemporal variability of tree water uptake using stable isotopes ($\delta$18O, $\delta$2H) in an alluvial system supplied by a high-altitude watershed, Pfyn forest, Switzerland, Ecohydrol., 7, 319–333,

doi:10.1002/eco.1347, 2014.

Brinkmann, N., Seeger, S., Weiler, M., Buchmann, N., Eugster, W., and Kahmen, A.: Employing stable isotopes to determine the residence times of soil water and the temporal origin of water taken up by Fagus sylvatica and Picea abies in a temperate forest, New Phytol, doi:10.1111/nph.15255, 2018.

Brooks, J. R., Barnard, H. R., Coulombe, R., and McDonnell, J. J.: Ecohydrologic separation of water between trees and streams in a Mediterranean climate, Nat Geosci, 3, 100–104, doi:10.1038/NGEO722, 2010.

Evaristo, J., Jasechko, S., and McDonnell, J. J.: Global separation of plant transpiration from groundwater and streamflow, Nature, 525, 91–94, doi:10.1038/nature14983, 2015.

Freyberg, J. von, Studer, B., and Kirchner, J. W.: A lab in the field: High-frequency analysis of water quality and stable isotopes in stream water and precipitation, Hydrol Earth Syst Sc, 21, 1721–1739, doi:10.5194/hess-21-1721-2017, 2017.

Geris, J., Tetzlaff, D., McDonnell, J., Anderson, J., Paton, G., and Soulsby, C.: Ecohydrological separation in wet, low energy northern environments?: A preliminary assessment using different soil water extraction techniques, Hydrol. Process., 29, 5139–5152, doi:10.1002/hyp.10603, 2015.

Gralher, B., Herbstritt, B., Weiler, M., Wassenaar, L. I., and Stumpp, C.: Correcting for Biogenic Gas Matrix Effects on Laser-Based Pore Water-Vapor Stable Isotope Measurements, Vadose Zone J, 17, 170157, doi:10.2136/vzj2017.08.0157, 2018.

Javaux, M., Rothfuss, Y., Vanderborght, J., Vereecken, H., and Bruggemann, N.: Isotopic composition of plant water sources, Nature, 536, E1-3, doi:10.1038/nature18946, 2016.

Knighton, J., Saia, S. M., Morris, C. K., Archiblad, J. A., and Walter, M. T.: Ecohydrologic Considerations for Modeling of Stable Water Isotopes in a Small Intermittent Watershed, Hydrol Process, 31, 2438–2452, doi:10.1002/hyp.11194, 2017.

Kuppel, S., Tetzlaff, D., Maneta, M. P., and Soulsby, C.: EcH2O-iso 1.0: Water isotopes and age tracking in a process-based, distributed ecohydrological model, Geosci.

Model Dev. Discuss., 1–38, doi:10.5194/gmd-2018-25, 2018.

Martín-Gómez, P., Serrano, L., Ferrio, J. P., and Cernusak, L.: Short-term dynamics of evaporative enrichment of xylem water in woody stems: Implications for ecohydrology, Tree Physiol, 1–12, doi:10.1093/treephys/tpw115, 2016.

McCutcheon, R. J., McNamara, J. P., Kohn, M. J., and Evans, S. L.: An Evaluation of the Ecohydrological Separation Hypothesis in a Semiarid Catchment, Hydrol. Process., 31, 783–799, doi:10.1002/hyp.11052, 2016.

Millar, C., Pratt, D., Schneider, D., and McDonnell, J. J.: A Comparison of Extraction Systems for Plant Water Stable Isotope Analysis, Rapid communications in mass spectrometry RCM, 32, 1031–1044, doi:10.1002/rcm.8136, 2018.

Orlowski, N., Pratt, D. L., and McDonnell, J. J.: Intercomparison of soil pore water extraction methods for stable isotope analysis, Hydrol. Process., 30, 3434–3449, doi:10.1002/hyp.10870, 2016.

Rothfuss, Y. and Javaux, M.: Reviews and syntheses: Isotopic approaches to quantify root water uptake: a review and comparison of methods, Biogeosciences, 14, 2199–2224, doi:10.5194/bg-14-2199-2017, 2017.

Scheliga, B., Tetzlaff, D., Nuetzmann, G., and Soulsby, C.: Groundwater isoscapes in a montane headwater catchment show dominance of well-mixed storage, Hydrol Process, 31, 3504–3519, doi:10.1002/hyp.11271, 2017.

Singer, M. B., Sargeant, C. I., Piégay, H., Riquier, J., Wilson, R. J. S., and Evans, C. M.: Floodplain ecohydrology: Climatic, anthropogenic, and local physical controls on partitioning of water sources to riparian trees, Water Resour Res, 50, 4490–4513, doi:10.1002/2014WR015581, 2014.

Smith, A. A., Welch, C., and Stadnyk, T. A.: Assessing the seasonality and uncertainty in evapotranspiration partitioning using a tracer-aided model, J Hydrol, 560, 595–613, doi:10.1016/j.jhydrol.2018.03.036, 2018.

Sprenger, M., Tetzlaff, D., Buttle, J., Laudon, H., and Soulsby, C.: Water ages in the critical zone of long-term experimental sites in northern latitudes, Hydrol. Earth Syst. Sci., (accepted), doi:10.5194/hess-2018-144, 2018a.

Sprenger, M., Tetzlaff, D., Buttle, J. M., Laudon, H., Leistert, H., Mitchell, C. P. J., Snelgrove, J., Weiler, M., and Soulsby, C.: Measuring and modelling stable isotopes of mobile and bulk soil water, Vadose Zone J, 17, 170149, doi:10.2136/VZJ2017.08.0149, 2018b.

Wang, L., Caylor, K. K., Villegas, J. C., Barron-Gafford, G. A., Breshears, D. D., and Huxman, T. E.: Partitioning evapotranspiration across gradients of woody plant cover: Assessment of a stable isotope technique, Geophys. Res. Lett., 37, L09401, doi:10.1029/2010GL043228, 2010.

Yang, J., Chen, H., Nie, Y., Zhang, W., and Wang, K.: Spatial variability of shallow soil moisture and its stable isotope values on a karst hillslope, Geoderma, 264, 61–70, doi:10.1016/j.geoderma.2015.10.003, 2016.

Zhao, L., Wang, L., Cernusak, L. A., Liu, X., Xiao, H., Zhou, M., and Zhang, S.: Significant Difference in Hydrogen Isotope Composition Between Xylem and Tissue Water in Populus Euphratica, Plant Cell Environ, 39, 1848–1857, doi:10.1111/pce.12753, 2016.

---

## Referee Comment (RC2) · S.P. Good (Referee) · 20 Jul 2018

Review of Penna et. al.:

The submitted paper is a review of recent studies in the use of stable isotopes to understand water cycling within ecosystems. The paper draws together multiple up-to-date papers that, when taken together, present a somewhat concerning view of the field as it currently stands. This I feel is much needed within the community, and publication of an even-handed review of these issues is well suited to Biogeoscicience. A few clarifications would be helpful before publication:

[Figure]

Wording: Given the above, it is critically important that the authors of this paper are very careful with their citations and the language used to discuss the conclusions of others. For example, on P5L10, the authors claim that in Brooks 2010, the sampled xylem water doesn't match the signature of potential source waters. I think this is a misinterpretation of the Brooks 2010 conclusions, which as stated in the abstract of that paper were that: "initial rainfall events after rainless summers is locked into small pores with low matric potential until transpiration empties these pores during following dry summers." So, Brooks 2010 et. al. do hypothesize that the xylem water does match the signature of a potential source, just not that of the well-integrated stream water. I do not take issue with your conclusion that the wrong pools are often sampled, just how this section (and others) is written could be improved for clairy and consistency. Another example is P5L24, where the authors state that soil water isotope ratios vary with water potential (citing Brantly 2017), then introduce the study of McCutcheon (2017) as a singular example that didn't find this to be to true. Given that you've got two competing recent phenomenological studies, some readers are likely to decided that this issues is decidedly unknown at this movement. However, reading through the paper as is currently worded, other readers could interpret that Penna and co-authors are definitively claiming that tight or loosely bound waters always have differing isotope ratios (though only one cation for this is given). Again, the wording needs to be clarified and more citations added.

Figures: A bit more work on your figure 2 would be helpful. I found the first two panels very difficult to interpret. I think a lot more thought needs to go into this figure and its description. For instance, your caption discusses multiple measurements made to characterize a heterogenous domain. However, panel A doesn't depict multiple samples, but only the variation within a sample as it's volume or duration increases. In panel B you show what looks like a gaussian distribution, but I fail to see how this represents the presence of both micro-scale and macro-scale heterogeneities. I find panel C the most informative here, but it also seems highly speculative and difficult to compare. Since you've got no x-axis, the location of the mean is meaningless, and all

you're really plotting is differences in standard deviations, so why not have similar set of axis as figure 1, but with variation itself? Also, what justification do you have for any of these standard deviations. This figure seems highly speculative and is only referenced in an off hand manner (P10L14). Finally, is figure 1 adapted from Bowen & Good 2015, it seems quite similar.

Spatial Patterns: On P7L7 the authors claim that less attention has been paid to spatial processes, which is decidedly not the case. Gabe Bowen has spent many years pursuing this topic in particular, published many papers, and even publish a book (Isoscapes) and recent review paper (Bowen and Good 2015). While much work has been focused on precipitation and surface water isotope ratios, many studies have extended this concept to plant and animal tissues.

---

## Referee Comment (RC3) · J. R. Brooks (Referee) · 7 Aug 2018

I read with great interest the commentary by Penna et al., as I wished I could have attended the meeting and was very interested in the discussions that ensued. Overall, I think this commentary will be a valuable summary of where we are in using water isotopes to partition stream-soil-plant interactions, and potential directions for the future. However, I feel that the urgent challenges outline here are often very vague in details, and ignore some work that has been published, giving the impression that we know a lot less than we actually do know. I think the impact would be much stronger if the authors gave more examples of the type of research they are suggesting to help

readers, and I provide some specific comments below on that. The authors also mischaracterize other studies or neglect some key pieces of work. These points can be easily addressed by the authors, and the result will be a valuable reflection on where we are in this field of research. Please see below for specific details.

First, since the audience of Biogeosciences Discussions is much broader than terrestrial ecology and critical zone studies, I think the title needs to read "Tracing terrestrial ecosystem water fluxes...". This commentary does not touch on the broad work of isotopic tracing within aquatic ecosystems. For the same reason, I think the authors should include a phrase that quickly defines critical zone, for those not working in the critical zone. Also, on P2, line 20, specify terrestrial ecosystems.

P2, L25: In mentioning advancements in isotope-based tools and methods, you should mention Sprenger et al. (2015) for an excellent summary.

P3, L13: You should probably include Gat (1996).

P3, L21: You should probably include Allen et al. (2017) when discussing evaporation from plant canopies.

P4, L5-7: When discussing the isotopic enrichment of heavy isotopes with leaf transpiration, the sentence on H and O exchange between $CO_2$ and $H_2O$ is really out of place. H does not exchange with $CO_2$, only oxygen, and the amount of oxygen in water is so vast that the oxygen in $CO_2$ does not really impact the water isotopic signature, and the authors don't provide a citation for their statement. Instead, well published processes such as the Péclet effect, and non-steady state processes are not even mentioned. There is a huge literature base on leaf water isotopes, but these sentences make it seem like a neglected area. See for example (Kahmen et al., 2008; Kahmen et al., 2009; Cernusak et al., 2016)

P4 L14-19: if you are going to compare IRMS and the laser techniques for measuring water, you should include the latest IAEA interlaboratory comparison of water isotopic

measures using both techniques (Wassenaar et al., 2018). They have a very interesting figure showing the problem with organics in water isotope analysis. This illustrates that even with this software, the problem is far from solved.

P4 Last line: include "terrestrial" before "ecosystems".

P5 L2: "How do plants select their water source?". This phrasing makes it seem like plants are consciously choosing their water sources. This section also seems to discount the vast literature by plant physiologist on plant water uptake through water potential gradients, and soil-plant continuum conceptual model. I don't believe any of the work with stable isotopes has refuted or made us question this conceptual model. See (Jackson et al., 2000). That said, while we know a lot about plant water uptake, I agree there are nuances we don't understand that the isotopic work has brought to light.

P5 L11: Please remove Brooks et al. 2010 from the reference list here referring to differences between plant and soil water. Brooks et al. (2010) focused on how soil water, particularly depleted soils at depth could be isotopically different from stream water, and that bulk soil water was different from lysimeter water collected at the same depth. The isotopic depletion found at depth could not be explained by evaporative processes.

P5 L24: I think the most appropriate reference for pore water extraction would be Sprenger et al. 2015. I don't recall McCutcheon et al. (2017) going into this issue, and I can't find it with a quick recheck of the paper.

P6 L3: Meinzer et al. (2006) showed it could take months.

P6, L16-19: I would say that other papers prior to this put forward these ideas.

P7, L16: You really need to include the work of Gabe Bowen when discussion spatial variation of precipitation isotopes, and any other spatial variation in water isotopes. See (Bowen & Revenaugh, 2003; Bowen, 2008).

P8 L4: Is this 2019 reference a typo?

P8 L11: You should include the work of Christine Stumpp here (Stumpp et al., 2007; Stumpp & Maloszewski, 2010).

When addressing heterogeneity within soil water, you gloss over general patterns we do see somewhat consistently. For example, that bound water shows more evaporative effects than lysimeter water collected at the same depth, and that bulk soil water isotopes generally decrease increasing soil depth. I think the section would be stronger if you did talk about these patterns.

P9, L5-6: I think saying "many trees have branches that are plumbed to specific roots" is misleading here. While not with isotopes, xylem transport with dyes and other tracers has been studied for a long time, and mostly mixing does occur, although not completely around the circumference, and it varies with xylem anatomy. For example, see Ellmore et al. (2006). Don't just highlight the extreme end of segmentation within plants, it's a continuum. It's likely only isotopically relevant for lateral vs tap roots.

P9 L9-10: Again, when talking about the spatial and temporal variation in leaf water fractionation processes, you state "this heterogeneity is often neglected.." but this variation has been the subject of many many studies. See my comment above, and many other leaf water papers out there including Helliker & Ehleringer (2002).

P9, L17-21: Please include more examples of work that reflects this across spatial scale work. For example, Sprenger et al. (2018) looked at the isotopic difference in lysimeter (mobile) and bulk water across a range of ecosystems. Brooks et al (2010) looked at 34 sites within one catchment to examine the spatial variation in soil water isotopes, but found depth explained more variation than location within the watershed such as ridge top vs riparian.

P9 L26-P10 L14: Please give examples here as to what you mean. I did not find figure 2 very helpful for these vague paragraphs. What you really mean here is what how good is the isotopic signal to noise ratio for the samples you are measuring. The signal is the variation across the scale of interest such as variance between sources. The

noise is the variance of repeated sampling of what is considered the same pool, such as xylem of multiple trees considered to be part of the same group. The signal should be multiple times greater than the noise. Experimental designs need to determine these variances. Variances generally decrease when samples integrate over larger space or time, but that is true for both the signal and the noise. Figure 2 kind of gets at that, but I felt it was confusing and not well explained.

P10, L23-30: I would also point out what Newberry et al. (2017) found about using oven dried soils, and our general method of testing the extraction protocol. I think it's important to highlight here. Also include Sprenger et al. (2015) review on pore water methodology.

P11, L4-18: These are very good points.

P11, L20-24: I agree this would be an exciting area to see researched in more detail. I think it would help readers if you gave a specific example of a physiological or ecohydrological process you would envision being aided by these techniques. Concrete examples help readers fully understand. Maybe expand on a labeling study, and explain how more high-resolution monitoring would have aided to more insights.

P12, L5-20: Again while a very important point, this paragraph is vague, and would be aided by more concrete examples.

P12, L21-29: I would go further here and say that studies need to do a better job of quantifying the variance within and between pools by duplicating every 10th or 20th sample. If your 10th sample is soil at 10 cm, collect two in the field, relatively near each other depending on study objectives.

P13 L14: Change to "natural and anthropogenic terrestrial enviornments. . .".

Overall, I look forward to seeing the authors develop this commentary further.

Allen ST, Keim RF, Barnard H, McDonnell JJ, Brooks JR. 2017. The role of stable isotopes in understanding rainfall interception processes: A review. WIREs Water 4:

[Figure]

e1187.

Bowen GJ. 2008. Spatial analysis of the intra-annual variation of precipitation isotope ratios and its climatological corollaries. Journal of Geophysical Research 113: D05113.

Bowen GJ, Revenaugh J. 2003. Interpolating the isotopic composition of modern meteoric precipitation. Water Resources Research DOI: 10.1029/2003WR002086

Brooks JR, Barnard H, Coulombe R, McDonnell JJ. 2010. Ecohydrologic separation of water between trees and streams in a Mediterranean climate. Nature Geoscience 3: 100-104.

Cernusak LA, Barbour MM, Arndt SK, Cheesman AW, English NB, Field TS, Helliker BR, Holloway-Phillips MM, Holtum JAM, Kahmen A, McInerney FA, Munksgaard NC, Simonin K, Song X, Stuart-Williams H, West JB, Farquhar GD. 2016. Stable isotopes in leaf water of terrestrial plants. Plant Cell and Environment 10.1111/pce.12703

Ellmore GS, Zanne AE, Orians CM. 2006. Comparative sectoriality in temperate hardwoods: hydraulics and xylem anatomy. Botanical Journal of the Linnean Society 150: 61-71. Gat J. 1996. Oxygen and hydrogen isotopes in the hydrologic cycle. Annual Review of Earth & Planetary Sciences 24: 225-262.

Helliker BR, Ehleringer JR. 2002. Grass blades as tree rings: environmentally induced changes in the oxygen isotope ratio of cellulose along the length of grass blades. New Phytologist 155: 417-424. Jackson RB, Sperry JS, Dawson TE. 2000. Root water uptake and transport: using physiological processes in global predictions. Trends in Plant Science 5: 482-488.

Kahmen A, Simmonin K, Tu KP, Goldsmith GR, Dawson TE. 2009. The influence of species and growing conditions on the 18-O enrichment of leaf water and its impact on 'effective path length'. New Phytologist 184: 619-630.

Kahmen A, Simonin K, Tu KP, Merchant A, Callister A, Siegwolf R, Dawson TE, Arndt SK. 2008. Effects of environmental parameters, leaf physiological properties and leaf

water relations on leaf water d18O enrichment in different Eucalyptus species. Plant, Cell and Environment 31: 738-751.

McCutcheon RJ, McNamara JP, Kohn MJ, Evans SL. 2017. An evaluation of the eco-hydrological separation hypothesis in a semiarid catchment. Hydrological Processes 31: 783-799.

Meinzer FC, Brooks JR, Domec JC, Gartner BL, Warren JM, Woodruff D, Bible K, Shaw DC. 2006. Dynamics of water transport and storage in conifers studied with deuterium and heat tracing techniques. Plant, Cell and Environment 29: 105-114.

Newberry SL, Nelson DB, Kahmen A. 2017. Cryogenic vacuum artifacts do not affect plant water‐uptake studies using stable isotope analysis. Ecohydrology 10: e1892.

Sprenger M, Herbstritt B, Weiler M. 2015. Established methods and new opportunities for pore water stable isotope analysis. Hydrological Processes 29: 5174-5192.

Sprenger M, Tetzlaff D, Buttle JM, Laudon H, Leistert H, Mitchell C, Snelgrove J, Weiler M, Soulsby C. 2018. Measuring and modelling stable isotopes of mobile and bulk soil water. Vadoze Zone Journal in press.

Stumpp C, Maloszewski P. 2010. Quantification of preferential flow and flow hetero-geneities in an unsaturated soil planted with different crops using the environmental isotope delta O-18. Journal of Hydrology 394: 407-415.

Stumpp C, Maloszewski P, Stichler W, Maciejewski S. 2007. Quantification of the heterogeneity of the unsaturated zone based on environmental deuterium observed in lysimeter experiments. Hydrological Sciences Journal-Journal Des Sciences Hydrologiques 52: 748-762.

Wassenaar L, Terzer-Wassmuth s, Douence C, Araguas-Araguas L, Aggarwal P, Coplen TB. 2018. Seeking excellence: An evaluation of 235 international laboratories conducting water isotope analyses by isotope‐ratio and laser‐absorption spectrometry. Rapid Communications in Mass Spectrometry 32: 393-406.

---

## Author Comment (AC1) · 20 Aug 2018

Response to Reviewer #1

"Tracing terrestrial ecosystem water fluxes using hydrogen and oxygen stable isotopes: challenges and opportunities from an interdisciplinary perspective", by D. Penna et al.

We thank Matthias Sprenger for the time he spent on our manuscript and his detailed comments that have helped us to improve the paper. The reviewer's comments are reproduced in their entirety and the authors' responses are given directly afterwards.

[Comment 1] I am happy to read that Daniele Penna and others attending the workshop

"Isotope based studies of water partitioning and plant-soil interactions in forested and agricultural environments" share the results of their discussions in the submitted commentary ("Ideas and perspectives" as called in Biogeosciences). They provide an adequate overview of the current state regarding recent developments in the application of stable isotopes as a tracer in biogeosciences and discuss also limitations that we are facing at the moment. I like how they frame these limitations as new perspectives and research opportunities to put a positive spin on these challenges. I also welcome their aim to promote interdisciplinary research using stable isotopes. The commentary is in the scope of the journal and I am sure it will be of interest to a broad community using stable isotopes. I suggest publication after minor revision and I provide some critical comments below. Please note that the provided references are of course only meant as suggestions and that reference to my own work are mainly provided to underline my arguments given in the general comments.

I think that the commentary could be stronger if the authors would revisit some parts, as the structure is not very consistent. For example, some parts in Section 2 would fit better to Section 3, as they deal with limitations, while several paragraphs of Section 3 read like a review and would fit better in Section 2. I provide example of these paragraphs in the specific comments.

[Response] We recognize that some improvements to the current manuscript structure are needed and that some sentences do not fit some sections perfectly. We will slightly modify the aim of Section 3 and will modify some lines accordingly in the revised version of the manuscript.

[Comment 2] I would also encourage the authors to acknowledge recent developments that deal with their suggested research agenda, when asking for isotopic variability of groundwater (as done by Scheliga et al., 2017), promoting tests of isotope analysis for vegetation samples (as done by Millar et al., 2018), analysis of spatial variability of soil water stable isotopes (as done by Yang et al., 2016), suggesting dual-labeling studies (as done by Bachmann et al., 2015), incorporating evaporation fractionation in

catchment models (as done by Knighton et al., 2017, Smith et al., 2018, and Kuppel et al., 2018). While these studies were partly published recently, I still think it would be worth looking into them and considering including them. This would provide examples of developments that might go into the direction that you are suggesting and help the reader to see what is currently being done and tested.

[Response] We thank Matthias Sprenger for suggesting these very recent publications that indeed fit well the topics addressed in our commentary. We will include these references in appropriate sections of the manuscript.

[Comment 3] One comment on the "n water world": The "Two Water Worlds" were suggested based on the findings by Brooks et al. (2010) that soil waters extracted with suction lysimeters or cryogenically have different isotopic compositions. Thus, this definition of a split into two subsurface pools mainly stems from the limitation to not be able to extract water held at different pressure heads. I obviously agree that the subsurface is a continuum of varying pressure heads (P6 L23). However, we are currently lacking the means to sample the isotopic composition along this continuum (water retention curve), but efforts have been made to compare different methods sampling waters of different mobility with different methods (see Figure 4 by Geris et al., 2015 and the work by Orlowski et al. (2016)). The relevance of different pressure heads for the TWW has been discussed earlier by Berry et al. (2017). The point that I want to raise here: Of course, it would be neat to be able to sample a "n water world", but is it really practical? Instead, the TWW is relatively loosely defined into "mobile" and "less mobile or tightly bound" waters. However, from a stable isotope perspective (on which the TWW is based), we can only distinguish between "mobile" and "bulk" soil water given our limitations in the sampling procedure with either suction lysimeter and cryogenic extraction/direct equilibration, respectively. I tried to convey this message with my co-authors in Sprenger et al. (2018b). Important with this regard is that i.) the relative contributions of "mobile" and "more tightly bound water" is temporally variable (Figure 5 in Sprenger et al., 2018b) and that ii.) the "mobile" water does not reflect at all the

total plant available water, but plants can access more tightly bound waters than the suction cup lysimeter (Figure 1 in Berry et al., 2017 and Figure 1 in Sprenger et al., 2018b). There is further a lack of clear definition distinguishing the two water pools.

[Response] We certainly agree with the Reviewer on this relevant point. Particularly, we recognize that there is a definitional problem with the term "water pools" because the definition is not univocal, and this hamper a clear conceptualization of the underlying physical and ecophysiological processes related to this. In the revised version of the manuscript, we will use the Reviewer's valuable suggestion to reformulate and extend the paragraph, also including the recommended references. Our edits will approximately be the following: "The proposed "two water worlds hypothesis" (McDonnell, 2014) has challenged the assumption of complete subsurface mixing that underlies many catchment models (Pfister and Kirchner, 2017). This hypothesis postulates that more mobile soil water contributes to groundwater recharge and streamflow whereas more tightly bound water tends to be used by plants (McDonnell, 2014; Evaristo et al., 2015). Preliminary evidence from catchment studies based on dual-isotope approach and conducted in different environments showed that bulk soil water was isotopically different from tension lysimeter water collected at the same depth (Brooks et al, 2010), that shallow soil water pool utilized by plants differed in isotopic composition from precipitation, stream baseflow, and soil-lysimeter water pools (Goldsmith et al., 2011), and that xylem water was isotopically similar to soil and rain water, but different from streamflow and groundwater (Penna et al., 2013). This conjecture has stimulated new interpretations of ecohydrological data and new research questions to investigate water flow pathways in catchments (McDonnell, 2014). However, there seems to be a trend to focus on the interpretation of recent data on just confirming or rejecting this one hypothesis. As outlined in Berry et al. (2017) and Sprenger et al. (2016), alternative hypotheses need to be developed and tested to improve our current understanding. Because water held in the soil or moving through the soil and other subsurface layers is a continuum, where water transport is driven by gradients, and not separate "worlds", we see the necessity to move from the simplistic "two water worlds hypothesis" to an "n

water worlds" concept, where multiple water reservoirs and flow pathways are invoked and parameterized, doing justice to the properties of the different substrate types and sites. However, in practical terms, we are currently lacking the means to sample the isotopic composition along this continuum, and we are limited to distinguish only between "mobile" and "bulk" soil water sampled by either tension lysimeters or cryogenic extraction and direct equilibration, respectively. Very recent findings highlighted that the relative contribution of mobile and more tightly bound water is temporally variable and that the mobile water does not reflect the total plant available water (Berry et al., 2017; Sprenger et al., 2018b). Although efforts have been made to compare different methods to sample water of different mobility for isotopic analysis (e.g., Geris et al., 2015; Orlowski et al., 2016b), we still lack a clear distinction of these two water pools. Investigating more in detail the relationships between soil water isotopic concentrations and water mobility at high spatial and temporal resolutions (McCutcheon et al., 2017) will help us to move beyond the two water worlds hypothesis."

Berry, Z. C., White, J. C. and Smith, W. K.: Foliar uptake, carbon fluxes and water status are affected by the timing of daily fog in saplings from a threatened cloud forest, Tree Physiol., 34(5), 459–470, doi:10.1093/treephys/tpu032, 2014.

Brooks, J. R., Barnard, H. R., Coulombe, R. and McDonnell, J. J.: Ecohydrologic separation of water between trees and streams in a Mediterranean climate, Nat. Geosci., 3(2), 100–104, doi:10.1038/ngeo722, 2010.

Evaristo, J., Jasechko, S. and McDonnell, J. J.: Global separation of plant transpiration from groundwater and streamflow, Nature, 525(7567), 91–94, doi:10.1038/nature14983, 2015.

Geris, J., Tetzlaff, D., Mcdonnell, J., Anderson, J., Paton, G. and Soulsby, C.: Ecohydrological separation in wet, low energy northern environments? A preliminary assessment using different soil water extraction techniques, Hydrol. Process., 29(25), 5139–5152, doi:10.1002/hyp.10603, 2015.

Goldsmith GR, Muñoz-Villers LE, Holwerda F, McDonnell JJ, Asbjornsen H, Dawson TE.: Stable isotopes reveal linkages among ecohydrological processes in a seasonally dry tropical montane cloud forest. Ecohydrology 2011, 5:779–790

McCutcheon, R. J., McNamara, J. P., Kohn, M. J. and Evans, S. L.: An evaluation of the ecohydrological separation hypothesis in a semiarid catchment, Hydrol. Process., 31(4), 783–799, doi:10.1002/hyp.11052, 2017.

McDonnell, J. J.: The two water worlds hypothesis: ecohydrological separation of water between streams and trees?, WIRES Water, 1, 323–329, doi:10.1002/wat2.1027, 2014.

Penna D, Oliviero O, Assendelft R, Zuecco G, Tromp-Meerveld I, Anfodillo T, Carraro V, Borga M, Dalla Fontana G. Tracing the water sources of trees and streams: isotopic analysis in a small pre-alpine catchment. Procedia Environ Sci 2013, 19:106–112

Orlowski, N., Pratt, D. L. and McDonnell, J. J.: Intercomparison of soil pore water extraction methods for stable isotope analysis, Hydrol. Process., 30(19), 3434–3449, doi:10.1002/hyp.10870, 2016b.

Pfister, L. and Kirchner, J. W.: Debates—Hypothesis testing in hydrology: Theory and practice, Water Resour. Res., 53(3), 1792–1798, doi:10.1002/2016WR020116, 2017.

Sprenger, M., Leistert, H., Gimbel, K. and Weiler, M.: Illuminating hydrological processes at the soil-vegetation-atmosphere interface with water stable isotopes, Rev. Geophys., 54(3), 674–704, doi:10.1002/2015RG000515, 2016.

Sprenger, M., Tetzlaff, D., Buttle, J. M., Laudon, H., Leistert, H., Mitchell, C. P. J., Snelgrove, J., Weiler, M., and Soulsby, C.: Measuring and modelling stable isotopes of mobile and bulk soil water, Vadose Zone J, 17, 170149, doi:10.2136/VZJ2017.08.0149, 2018b.

[Comment 4] P2 L11: I am not sure if a report (2030 WRG, 2009) sponsored by "The Barilla Group, The Coca-Cola Company, The International Finance Corporation, McK-

insey Company, Nestlé S.A., New Holland Agriculture, SABMiller plc, Standard Chartered Bank, and Syngenta AG" is an appropriate reference.

[Response] Indeed it is not! It was a slip and we will remove it.

[Comment 5] P2 L 29: To my understanding, it is not only that the laser-based instruments are more affordable, but also running it for analysis is cheaper and easier.

[Response] That is true. We will specify this in the revised manuscript.

[Comment 6] P3 L2: I think that the context is missing here. One might ask "which well-mixed conditions"? While I believe to know what you refer to, you aim to reach a wider audience in Biogeosciences and thus, should be clear about the simplifying assumptions.

[Response] We agree with the reviewer. We will modify the sentence as follows: "Consequently, recent studies have revealed problems in the simplifying assumptions that underlie past investigations, especially those related to steady-state and well-mixed conditions, i.e., the assumption that water in the subsurface mixes instantaneously and completely in one common reservoir, so that no differences in isotopic composition would be observed with depth and in space.

[Comment 7] P3 L25-L29: This reads more like "limitations and challenges" and is an example of what I mean that the structure could be improved.

[Response] We agree, and we will move this paragraph to Section 3.2, when explaining sources of heterogeneity in the isotopic composition of soil water.

[Comment 8] P4 L17-L19: Again, when you start a sentence with "One limitation: : :", this might fit better in section 3 of the manuscript.

[Response] We will reword this sentence to present the characteristics of laser instruments.

[Comment 9] P4 L19: What about carrier gas issues like CO2, which potentially cause

issues for CRDS (Gralher et al., 2018)? Beside consequences for the direct-vapor equilibration method, this would also be relevant for in-situ measurements via the vapor phase.

[Response] Yes, this is a relevant issue, and we will add another statement in this paragraph.

[Comment 10] P4 L23: You mention soils, tree stems and leaf as examples. What about ETpartitioning as done for example by Wang et al. (2010)?

[Response] Even though this is not a recent study, we will add this reference.

[Comment 11] P5 L6: I think that the comparison and review study by Rothfuss and Javaux (2017) is a more appropriate reference here.

[Response] We will add this reference.

[Comment 12] P5 L10: Given the methodological issues with the calculations of source waters by Evaristo et al. (2015), as revealed by Javaux et al. (2016), this might not be a good reference.

[Response] Both Javaux et al. (2016), in their comment, and Evaristo et al. (2016) in their reply recognize that the mistake did not affect the central conclusion of the paper. Therefore, we believe that the reference is still appropriate and we will keep it in the manuscript.

Evaristo, J., Jasechko, S., McDonnell, J.J., 2016. Evaristo et al. reply. Nature 536, E3.

Javaux, M., Rothfuss, Y., Vanderborght, J., Vereecken, H., Brüggemann, N., 2016. Isotopic composition of plant water sources. Nature 536, E1.

[Comment 13] P5 L18: Consider including Zhao et al. (2016) as reference, who compared also different methods for xylem sampling.

[Response] Yes, this reference was already cited elsewhere in the original manuscript,

and we will add a citation also here.

[Comment 14] P5 L20: Be more specific: What do you suggest to the community? What do you "aim to analyze" (- depends on your research question, or not)? How is this message different to the section 4.3 in the commentary by Berry et al. (2017)?

[Response] We believe that we cannot be more specific because specific suggestions depend on the research question of individual studies. Therefore, we are suggesting to focus on two main issues already mentioned by Berry et al. (2017), somehow re-iterating their message. We will include a further citation to the work of Berry et al. (2017).

Berry, Z. C., Evaristo, J., Moore, G., Poca, M., Steppe, K., Verrot, L., Asbjornsen, H., Borma, L. S., Bretfeld, M., Hervé-Fernández, P., Seyfried, M., Schwendenmann, L., Sinacore, K., De Wispelaere, L. and Mcdonnell, J.: The two water worlds hypothesis: Addressing multiple working hypotheses and proposing a way forward, Ecohydrology, 1–10, doi:10.1002/eco.1843, 2017.

[Comment 15] P5 L26: I am not aware that McCutcheon et al. (2016) studied the stable isotopic compositions among different pore spaces. Instead, they specifically stated that "We are not able to determine if pore-scale variability can impact isotopic composition of root-absorbed and draining water".

[Response] True, that reference is not appropriate there, and we will remove it.

[Comment 16] P5 L31: How about evaporation through the bark as studied by Martín-Gómez et al. (2016)?

[Response] Yes, we will add these findings and the reference.

[Comment 17] P6 L7: Do you think numerical simulations can help assessing such ages, as done by Brinkmann et al. (2018) and Sprenger et al. (2018a)?

[Response] These are relevant studies in this context. We will modify the sentence

including results from modeling studies.

[Comment 18] P6 L14: Do you think relating tree ring to source waters would be beneficial to study long term dynamics (e.g., Singer et al., 2014)?

[Response] In this paragraph, we discuss the short-term variability of the isotopic composition of water sources. The tree ring timespan goes well beyond this timescale. In addition, the determination of the water source is not straightforward because many processes influence the signature of cellulose, which is only partially related to that of the source water (Gessler et al 2014). Therefore, we think that it is not appropriate to add the suggested reference in this paragraph. However, we will include it in Section 3.2, when discussing wood isotopic composition.

Gessler, A., Ferrio, J. P., Hommel, R., Treydte, K., Werner, R. A. and Monson, R. K.: Stable isotopes in tree rings: towards a mechanistic understanding of isotope fractionation and mixing processes from the leaves to the wood, Tree Physiol., 34(8), 796–818, doi:10.1093/treephys/tpu040, 2014.

[Comment 19] P6 L25: See general comments regarding the "n Water World".

[Response] Please, see our response to comment 3 above.

[Comment 20] P7 L18- P8 L23: These paragraphs read like a review and do not link to "Limitations and challenges".

[Response] We will slightly restructure the manuscript focusing the entire Section 3 on summarizing the current knowledge about the source of variability in the isotopic composition of the different ecohydrological compartments, as well as highlighting the limitations and challenges deriving from the heterogeneity of the systems of interest and the uncertainty associated to our current knowledge and measurements efforts. Moreover, we will reword some sentences to highlight the difficulties of studying ecohydrological systems and the related challenges.

[Comment 21] P8 L26: Please note the intensive sampling of groundwater stable iso-

topes by Scheliga et al. (2017).

[Response] Thanks. We will rephrase the sentence and refer to this study.

[Comment 22] P8 L29- P9 L10: To me, you miss pointing out the limitations and challenges in these paragraphs.

[Response] Please, see our response to comment 20.

[Comment 23] P9 L21: Regarding the heterogeneity of soil water stable isotopes, please see the data set by Yang et al. (2016) the intense variability across relatively small area.

[Response] The reference is relevant to highlight the spatial variability of the isotopic composition of soil water, depending on different factors. However, in this sentence, we are referring to the heterogeneity in the isotopic composition of water when moving up in scale in more general terms of ecohydrological processes than soil water processes only. However, we will include the suggested reference in two points of Section 3.2 of the revised manuscript, when mentioning the effect of the isotopic composition of antecedent rainfall on soil water, as well as the effect of local soil properties.

[Comment 24] P10 L23: Consider including the interesting study by Millar et al. (2018)

[Response] Yes, we will add this citation, as it is highly relevant.

[Comment 25] P10 L14: This ratio between water accessible for plants and water extractable by suction lysimeter is time variant (Sprenger et al., 2018b) – this is usually ignored and cannot be accounted for when sampling twice soil and plant isotopes over a year.

[Response] The reviewer likely refers to P11 L14. This is a very relevant study, published online only a few days before we submitted our commentary. We will include the point raised by the reviewer in the revised manuscript.

[Comment 26] P12 L5: I suggest providing examples, where this has been done (e.g.,

Knighton et al., 2017, Smith et al., 2018, and Kuppel et al., 2018)

[Response] Yes, we will do that.

[Comment 27] P12 L1: As done by Bachmann et al. (2015)

[Response] We will add this reference.

[Comment 28] P12 L12: as reviewed by Rothfuss and Javaux (2017)

[Response] We will add the following sentences to the manuscript: "For instance, Rothfuss and Javaux (2017) examined the uncertainty associated to different types of mixing models, stating that graphical and statistical methods have major drawbacks when analyzing root water uptake depths. They found that the latest generation of Bayesian mixing models performs well for that purpose, but only when the number of considered water sources in the soil is high and closely reflects the vertical distribution of the soil water isotopic composition."

Rothfuss, Y. and Javaux, M.: Reviews and syntheses: Isotopic approaches to quantify root water uptake: a review and comparison of methods, Biogeosciences, 14, 2199–2224, doi:10.5194/bg-14-2199-2017, 2017

[Comment 29] P12 L14: Can you provide any studies backing this idea? Is to "introduce further complexity" something positive in this context? This "potential interactions of these tracers with soil, roots and the water itself" would need to be carefully studied and understood for the tracer interpretations, right?

[Response] We agree and we will reformulate as follows: "Additional tracers can be helpful to support and strengthen the observations obtained by using stable isotopes. Trace elements taken up through plants (e.g., gold particles, Lintern et al., 2013) might have particular potential for inferring root water uptake. Also, chemical tracers (e.g., Haase et al., 1996) and tritium (Zhang et al., 2017) have been used to study water uptake depths. However, using other tracers such as fluorobenzoic acids, dissolved ions, isotopic ratios of other elements such as radium or strontium, will introduce further complexity to the system due to potential interactions of these tracers with soil, roots and the water itself. Such effects need to be carefully studied in order to provide meaningful interpretations."

Haase, P., Pugnaire, F.I., Fernández, E.M., Puigdefábregas, J., Clark, S.C., Incoll, L.D., an investigation of rooting depth of the semiarid shrub Retama sphaerocarpa (L.) Boiss. by labelling of ground water with a chemical tracer. J. Hydrol. 177, 23–31, 1996. https://doi.org/10.1016/0022-1694(95)02794-7

Lintern, M., Anand, R., Ryan, C., Paterson, D.,Natural gold particles in Eucalyptus leaves and their relevance to exploration for buried gold deposits, Nat. Commun. 4, 2274, 2013. https://doi.org/10.1038/ncomms3614

Zhang ZQ, Evaristo J, Li Z, Si BC, McDonnell JJ. Tritium analysis shows apple trees may be transpiring water several decades old. Hydrological Processes. 2017;31:1196–1201. https://doi.org/10.1002/hyp.11108

[Comment 30] Fig. 1: You've added remote sensing to the graph, but do not discuss it in the text.

[Response] We will include more discussion on remote sensing in the text.

[Comment 31] Fig. 1: Plant uptake and transpiration have scales > 10m in your graph, but much of your discussion deals with processes way smaller where water is taken up by roots (scales Âń 1m).

[Response] This is a good point. We will modify the figure to highlight the processes of root uptake and leaf transpiration at the scale of individuals (1 m and below) and that of including vegetation processes at larger scales (stands, forests, agroecosystems, etc).

[Comment 32] Fig. 1: How about potential of tree ring samples (in the category "grab sampling")?

[Response] As we have not discussed this topic in the manuscript, we prefer to leave it

out from this conceptual sketch.

[Comment 33] Fig. 2A: It seems from Fig. 2B that the green shading represents the range of samples and the black line represents an average value. Are these values then representing average and range from replicates (e.g., five soil samples at 0-5 cm soil depth)? Otherwise, how could you have a range from one sample value (as indicated on the y-axis)? Or does the green shade the range of values out there in nature and the black line represents the individual sample? This is not clear to me.

Fig. 2C: Not sure if I understand how the temporal variability of precipitation and throughfall is higher within events than among events. Is not the variability within the events part of the overall variability among events? Looking at Figure 6 in Freyberg et al. (2017) it seems that variability between events is higher.

Fig. 2C: Why did you not include plant water isotopes?

[Response] The inclusion of this figure in the manuscript stemmed from debates engaged in the discussion groups at the workshop. Introducing this figure aimed at providing a concept of a representative sampling size or scale over which to bulk a sample. It did not focus on averaging multiple samples but on achieving a bulk sample of a typical size that could reflect – upon multiple samplings – the variability (distribution of mean and standard variation) that is representative for the sample type (soil water, plant water etc). Ultimately, these considerations would result into the recommendation of typical sample sizes and frequencies. However, we share the perplexities of all three reviewers on the usefulness of this figure, and we agree with them that the figure is vague and may be confusing. In addition, it is not strictly related to the text and the main focus of this commentary. Therefore, we will remove this figure from the revised version of the manuscript, as well as mention to it in the text.

[Comment 34] Title: As I understand, the manuscript title must start with "Ideas and perspectives:" (https://www.biogeosciences.net/about/manuscript_types.html)

[Response] True, we will implement that in the revised manuscript.

[Comment 35] P2 L24: Repetition of "Stable isotopes of hydrogen and oxygen"

[Response] We will modify the sentence to avoid the repetition.

[Comment 36] P14 L1: Coauthors missing for Bertrand et al. (2014).

[Response] True, we will add the complete reference.

---

## Author Comment (AC2) · 20 Aug 2018

Response to Reviewer #2

"Tracing terrestrial ecosystem water fluxes using hydrogen and oxygen stable isotopes: challenges and opportunities from an interdisciplinary perspective", by D. Penna et al.

We thank dr. Steve Good for the very useful comments he gave on our manuscript that have helped us to improve the paper. The reviewer's comments are reproduced in their entirety and the authors' responses are given directly afterward.

[Comment 1] The submitted paper is a review of recent studies in the use of stable

isotopes to understand water cycling within ecosystems. The paper draws together multiple up-to date papers that, when taken together, present a somewhat concerning view of the field as it currently stands. This I feel is much needed within the community, and publication of an even-handed review of these issues is well suited to Biogeoscicience. A few clarifications would be helpful before publication. Wording: Given the above, it is critically important that the authors of this paper are very careful with their citations and the language used to discuss the conclusions of others. For example, on P5L10, the authors claim that in Brooks 2010, the sampled xylem water doesn't match the signature of potential source waters. I think this is a misinterpretation of the Brooks 2010 conclusions, which as stated in the abstract of that paper were that: "initial rainfall events after rainless summers is locked into small pores with low matric potential until transpiration empties these pores during following dry summers." So, Brooks 2010 et. al. do hypothesize that the xylem water does match the signature of a potential source, just not that of the well-integrated stream water. I do not take issue with your conclusion that the wrong pools are often sampled, just how this section (and others) is written could be improved for clarity and consistency.

[Response] We thank the reviewer for noticing this inappropriate interpretation of the findings by Brooks et al (2010). We will modify the sentence accordingly.

[Comment 2] Another example is P5L24, where the authors state that soil water isotope ratios vary with water potential (citing Brantly 2017), then introduce the study of McCutcheon (2017) as a singular example that didn't find this to be to true. Given that you have two competing recent phenomenological studies, some readers are likely to decide that this issue is decidedly unknown at this movement. However, reading through the paper as is currently worded, other readers could interpret that Penna and co-authors are definitively claiming that tight or loosely bound waters always have differing isotope ratios (though only one cation for this is given). Again, the wording needs to be clarified and more citations added.

[Response] We will clarify the sentence reporting the correct citations and presenting

this concept more clearly.

[Comment 3] Figures: A bit more work on your figure 2 would be helpful. I found the first two panels very difficult to interpret. I think a lot more thought needs to go into this figure and its description. For instance, your caption discusses multiple measurements made to characterize a heterogeneous domain. However, panel A doesn't depict multiple samples, but only the variation within a sample as it's volume or duration increases. In panel B you show what looks like a gaussian distribution, but I fail to see how this represents the presence of both micro-scale and macro-scale heterogeneities. I find panel C the most informative here, but it also seems highly speculative and difficult to compare. Since you've got no x-axis, the location of the mean is meaningless, and all you're really plotting is differences in standard deviations, so why not have similar set of axis as figure 1, but with variation itself? Also, what justification do you have for any of these standard deviations. This figure seems highly speculative and is only referenced in an off hand manner (P10L14).

[Response] This figure will be removed from the revised manuscript. As reported in the response to comment 33 by Reviewer 1, the inclusion of this figure in the manuscript stemmed from debates engaged in the discussion groups at the workshop. Introducing this figure aimed at providing a concept of a representative sampling size or scale over which to bulk a sample. It did not focus on averaging multiple samples but on achieving a bulk sample of a typical size that could reflect – upon multiple samplings – the variability (distribution of mean and standard variation) that is representative for the sample type (soil water, plant water etc). Ultimately, these considerations would result into the recommendation of typical sample sizes and frequencies. However, we share the perplexities of all three reviewers on the usefulness of this figure, and we agree with them that the figure is vague and may be confusing. In addition, it is not strictly related to the text and the main focus of this commentary. Therefore, we will remove this figure from the revised version of the manuscript, as well as mention to it in the text.

[Comment 4] Finally, is figure 1 adapted from Bowen & Good 2015, it seems quite similar.

[Response] We apologize for this. The figure is an original work but based on Blöschl and Sivapalan (1995), and Bowen and Good (2015). We will specify this in the revised version.

Blöschl, G. and Sivapalan, M., Scale issues in hydrological modelling: a review, Hydrol Process, 9(3-4), 251-290, doi: 10.1002/hyp.3360090305, 1995.

Bowen, G. and Good S.P., Incorporating water isoscapes in hydrological and water resource investigations, WIREs Water 2015, 2:107–119. doi: 10.1002/wat2.1069

[Comment 5] Spatial Patterns: On P7L7 the authors claim that less attention has been paid to spatial processes, which is decidedly not the case. Gabe Bowen has spent many years pursuing this topic in particular, published many papers, and even publish a book (Isoscapes) and recent review paper (Bowen and Good 2015). While much work has been focused on precipitation and surface water isotope ratios, many studies have extended this concept to plant and animal tissues.

[Response] We are aware of the relevant work by Gabriel Bowen on isoscapes and spatial patterns of isotopic composition in hydrological systems. Here, we are mainly referring to the small spatial scales, especially for soil water and groundwater isotopic composition, while isoscapes are more traditionally applied at larger scales. However, the topic is appropriate and we will re-arrange the sentence and include the concept of isoscapes.
* * *

---

## Author Comment (AC3) · 20 Aug 2018

Response to Reviewer #3

"Tracing terrestrial ecosystem water fluxes using hydrogen and oxygen stable isotopes: challenges and opportunities from an interdisciplinary perspective", by D. Penna et al.

We thank Dr. Renee Brooks for her suggestions, corrections, and recommendations on our manuscript that have helped us to improve the paper. The reviewer's comments are reproduced in their entirety and the authors' responses are given directly afterward.

[Comment 1] I read with great interest the commentary by Penna et al., as I wished

[Figure]

I could have attended the meeting and was very interested in the discussions that ensued. Overall, I think this commentary will be a valuable summary of where we are in using water isotopes to partition stream-soil-plant interactions, and potential directions for the future. However, I feel that the urgent challenges outline here are often very vague in details, and ignore some work that has been published, giving the impression that we know a lot less than we actually do know. I think the impact would be much stronger if the authors gave more examples of the type of research they are suggesting to help readers, and I provide some specific comments below on that. The authors also mischaracterize other studies or neglect some key pieces of work. These points can be easily addressed by the authors, and the result will be a valuable reflection on where we are in this field of research. Please see below for specific details.

[Response] We thank the reviewer for highlighting the strong points and, at the same time, for outlining the weak points of our commentary, as well as for suggesting other key literature studies that were missing. Please, see the response to the specific comments below for the details on how we will modify the manuscript to take these indications into account.

[Comment 2] First, since the audience of Biogeosciences Discussions is much broader than terrestrial ecology and critical zone studies, I think the title needs to read "Tracing terrestrial ecosystem water fluxes: : :". This commentary does not touch on the broad work of isotopic tracing within aquatic ecosystems. For the same reason, I think the authors should include a phrase that quickly defines critical zone, for those not working in the critical zone. Also, on P2, line 20, specify terrestrial ecosystems.

[Response] We agree with this suggestion, and we will change the title and the definition of ecosystems as recommended. The traditional definition of the critical zone is given at the very beginning of Section 1.

[Comment 3] P2, L25: In mentioning advancements in isotope-based tools and methods, you should mention Sprenger et al. (2015) for an excellent summary.

[Figure]

[Response] Yes, we will.

[Comment 4] P3, L13: You should probably include Gat (1996).

[Response] Yes, we will.

[Comment 5] P3, L21: You should probably include Allen et al. (2017) when discussing evaporation from plant canopies.

[Response] That paper is cited elsewhere in the manuscript but it certainly fits here as well. We will add it.

[Comment 6] P4, L5-7: When discussing the isotopic enrichment of heavy isotopes with leaf transpiration, the sentence on H and O exchange between $CO_2$ and $H_2O$ is really out of place. H does not exchange with $CO_2$, only oxygen, and the amount of oxygen in water is so vast that the oxygen in $CO_2$ does not really impact the water isotopic signature, and the authors don't provide a citation for their statement. Instead, well published processes such as the Péclet effect, and non-steady state processes are not even mentioned. There is a huge literature base on leaf water isotopes, but these sentences make it seem like a neglected area. See for example (Kahmen et al., 2008; Kahmen et al., 2009; Cernusak et al., 2016)

[Response] We thank the reviewer for noticing this issue. Mentioning and explaining, even shortly, processes such as H and O exchange, Péclet effect, and non-steady state processes would require too much space and would be out of the scope of this commentary. So, we prefer to remove the sentence that was out of place.

[Comment 7] P4 L14-19: if you are going to compare IRMS and the laser techniques for measuring water, you should include the latest IAEA interlaboratory comparison of water isotopic measures using both techniques (Wassenaar et al., 2018). They have a very interesting figure showing the problem with organics in water isotope analysis. This illustrates that even with this software, the problem is far from solved.

[Response] We were aware of the WICO2018 intercomparison test, and of the published results. We will add this reference to report that the organic contamination on water isotopic measurements is still an unsolved problem.

[Comment 8] P4 Last line: include "terrestrial" before "ecosystems".

[Response] We will.

[Comment 9] P5 L2: "How do plants select their water source?". This phrasing makes it seem like plants are consciously choosing their water sources. This section also seems to discount the vast literature by plant physiologist on plant water uptake through water potential gradients, and soil-plant continuum conceptual model. I don't believe any of the work with stable isotopes has refuted or made us question this conceptual model. See (Jackson et al., 2000). That said, while we know a lot about plant water uptake, I agree there are nuances we don't understand that the isotopic work has brought to light.

[Response] We will reformulate this question to eliminate any possible confusion about the consciousness of plants in selecting water sources. We will also make it clear that, in this section, we do not intend to discuss the physical mechanisms that drive water from the soil, to the plant, and to the atmosphere. We are aware of the suggested paper (Jackson et al., 2000) and of the vast knowledge already at our disposal on this topic but we believe that discussing or summarizing these findings goes beyond the aim of this commentary.

[Comment 10] P5 L11: Please remove Brooks et al. 2010 from the reference list here referring to differences between plant and soil water. Brooks et al. (2010) focused on how soil water, particularly depleted soils at depth could be isotopically different from stream water, and that bulk soil water was different from lysimeter water collected at the same depth. The isotopic depletion found at depth could not be explained by evaporative processes.

[Response] We completely agree and apologize to the reviewer for having misinterpreted the data of this benchmark work. As replied to the first comment of Reviewer 2, we will substantially modify the sentence so to correct this relevant point.

[Comment 11] P5 L24: I think the most appropriate reference for pore water extraction would be Sprenger et al. 2015. I don't recall McCutcheon et al. (2017) going into this issue, and I can't find it with a quick recheck of the paper.

[Response] We agree and will replace the reference with the suggested one.

[Comment 12] P6 L3: Meinzer et al. (2006) showed it could take months.

[Response] Thanks, we will add this.

[Comment 13] P6, L16-19: I would say that other papers prior to this put forward these ideas.

[Response] We will reformulate the sentence and add references that showed first evidences of these ideas.

[Comment 14] P7, L16: You really need to include the work of Gabe Bowen when discussion spatial variation of precipitation isotopes, and any other spatial variation in water isotopes. See (Bowen & Revenaugh, 2003; Bowen, 2008).

[Response] True. We will include these references. We will also include a reference to Bowen and Good (2015) and to the isosocape concept in the revised Section 3.2.

[Comment 15] P8 L4: Is this 2019 reference a typo?

[Response] No, it is not a typo. That paper is available online but it will appear in a special issue that will be published in 2019. As reported by the journal, the publication year of this article is 2019.

[Comment 16] P8 L11: You should include the work of Christine Stumpp here (Stumpp et al., 2007; Stumpp & Maloszewski, 2010).

[Response] We will consider these as well as other appropriate references here.

[Comment 17] When addressing heterogeneity within soil water, you gloss over general patterns we do see somewhat consistently. For example, that bound water shows more evaporative effects than lysimeter water collected at the same depth, and that bulk soil water isotopes generally decrease increasing soil depth. I think the section would be stronger if you did talk about these patterns.

[Response] We will add a new paragraph in this section discussing these patterns and citing relevant recent studies.

[Comment 18] P9, L5-6: I think saying "many trees have branches that are plumbed to specific roots" is misleading here. While not with isotopes, xylem transport with dyes and other tracers has been studied for a long time, and mostly mixing does occur, although not completely around the circumference, and it varies with xylem anatomy. For example, see Ellmore et al. (2006). Don't just highlight the extreme end of segmentation within plants, it's a continuum. It's likely only isotopically relevant for lateral vs tap roots.

[Response] We will modify the sentence to avoid misleading information.

[Comment 19] P9 L9-10: Again, when talking about the spatial and temporal variation in leaf water fractionation processes, you state "this heterogeneity is often neglected.." but this variation has been the subject of many many studies. See my comment above, and many other leaf water papers out there including Helliker & Ehleringer (2002).

[Response] We will modify the sentence as follows: "These fractionation effects are spatio-temporally variable (see, for instance, Helliker and Ehleringer, 2002), although possibly masked by wood and other tissues that might act as temporal and spatial integrators of heterogeneous processes in leaves (Gessler et al., 2014; Singer et al., 2014)."

Helliker BR, Ehleringer JR.: Grass blades as tree rings: environmentally induced changes in the oxygen isotope ratio of cellulose along the length of grass blades. New

Phytologist 155: 417-424, 2002.

Gessler, A., Ferrio, J. P., Hommel, R., Treydte, K., Werner, R. A. and Monson, R. K.: Stable isotopes in tree rings: towards a mechanistic understanding of isotope fractionation and mixing processes from the leaves to the wood, Tree Physiol., 34(8), 796–818, doi:10.1093/treephys/tpu040, 2014.

Singer, M. B., Sargeant, C. I., Piégay, H., Riquier, J., Wilson, R. J. S., and Evans, C. M.: Floodplain ecohydrology: Climatic, anthropogenic, and local physical controls on partitioning of water sources to riparian trees, Water Resour Res, 50, 4490–4513, doi:10.1002/2014WR015581, 2014

[Comment 20] P9, L17-21: Please include more examples of work that reflects this across spatial scale work. For example, Sprenger et al. (2018) looked at the isotopic difference in lysimeter (mobile) and bulk water across a range of ecosystems. Brooks et al (2010) looked at 34 sites within one catchment to examine the spatial variation in soil water isotopes, but found depth explained more variation than location within the watershed such as ridge top vs riparian.

[Response] In the revised manuscript, we will include five additional examples from recent (2016-2018) studies on this topic.

[Comment 21] P9 L26-P10 L14: Please give examples here as to what you mean. I did not find figure 2 very helpful for these vague paragraphs. What you really mean here is what how good is the isotopic signal to noise ratio for the samples you are measuring. The signal is the variation across the scale of interest such as variance between sources. The noise is the variance of repeated sampling of what is considered the same pool, such as xylem of multiple trees considered to be part of the same group. The signal should be multiple times greater than the noise. Experimental designs need to determine these variances. Variances generally decrease when samples integrate over larger space or time, but that is true for both the signal and the noise. Figure 2 kind of gets at that, but I felt it was confusing and not well explained.

[Response] Thank you for this useful suggestion that we will integrate in the revised explanation of this concept. We will also try to give some practical examples to allow the reader to better understand our message. Finally, Fig. 2 will be removed from the revised manuscript. Please, see our response to comment 33 by Reviewer 1 in this regard.

[Comment 22] P10, L23-30: I would also point out what Newberry et al. (2017) found about using oven dried soils, and our general method of testing the extraction protocol. I think it's important to highlight here. Also include Sprenger et al. (2015) review on pore water methodology.

[Response] We will change the statement accordingly.

[Comment 23] P11, L4-18: These are very good points.

[Response] Thanks!

[Comment 24] P11, L20-24: I agree this would be an exciting area to see researched in more detail. I think it would help readers if you gave a specific example of a physiological or ecohydrological process you would envision being aided by these techniques. Concrete examples help readers fully understand. Maybe expand on a labeling study, and explain how more high-resolution monitoring would have aided to more insights.

[Response] We will expand the section giving examples of processes we think will benefit from high-resolution monitoring and/or labeling experiments.

[Comment 25] P12, L5-20: Again while a very important point, this paragraph is vague, and would be aided by more concrete examples.

[Response] Also following the suggestions of the two other reviewers, we will add examples to make this paragraph more specific.

[Comment 26] P12, L21-29: I would go further here and say that studies need to do a better job of quantifying the variance within and between pools by duplicating every

10th or 20th sample. If your 10th sample is soil at 10 cm, collect two in the field, relatively near each other depending on study objectives.

[Response] We will add a sentence indicating that.

[Comment 27] P13 L14: Change to "natural and anthropogenic terrestrial environments.

[Response] We will do it.

---

## Author Response (AR1)

**Response to the Editor**

**"Ideas and perspectives: Tracing terrestrial ecosystem water fluxes using hydrogen and oxygen stable isotopes - challenges and opportunities from an interdisciplinary perspective", by D. Penna et al.**

Dear Editor,

We are submitting the revised version of our manuscript entitled "Ideas and perspectives: Tracing terrestrial ecosystem water fluxes using hydrogen and oxygen stable isotopes - challenges and opportunities from an interdisciplinary perspective".

We have made all edits and modifications described in our point-by-point response to the three reviewers, including expansion of discussion on some critical points, a significant addition of new references, corrections of wrong statements, clarification of confused sentences, changes in the first figure and removal of the second one.

Moreover, as suggested, we added discussion about the relevant issue of plant use of tightly bound soil water, and have included our view on this topic, stressing the need for new interdisciplinary work to address this conundrum more in detail.

We feel that addressing the points raised by the reviewers and by the Editor significantly improved our manuscript and made it of greater impact on the readers. Given the overall positive evaluation provided by all three reviewers and the Editor, we hope that our revised manuscript can be directly evaluated by the Editor. We believe that fast publication is important to keep the discussion alive and to stimulate further discussion in the scientific community – one of the main reasons why we chose this journal for our paper.

Thank you very much for your support.

Warm regards,

Dr. Francesca Scandellari
on behalf of the authors of this manuscript

[revised manuscript text omitted]

.